# BLOCK DIFFUSION: INTERPOLATING BETWEEN AUTOREGRESSIVE AND DIFFUSION LANGUAGE MODELS

**Marianne Arriola**[†] [*]     **Aaron Kerem Gokaslan**[†]     **Justin T. Chiu**[‡]     **Zhihan Yang**[†]

**Zhixuan Qi**[†]     **Jiaqi Han**[¶]     **Subham Sekhar Sahoo**[†]     **Volodymyr Kuleshov**[†]

## ABSTRACT

Diffusion language models offer unique benefits over autoregressive models due to their potential for parallelized generation and controllability, yet they lag in likelihood modeling and are limited to fixed-length generation. In this work, we introduce a class of block diffusion language models that interpolate between discrete denoising diffusion and autoregressive models. Block diffusion overcomes key limitations of both approaches by supporting flexible-length generation and improving inference efficiency with KV caching and parallel token sampling. We propose a recipe for building effective block diffusion models that includes an efficient training algorithm, estimators of gradient variance, and data-driven noise schedules to minimize the variance. Block diffusion sets a new state-of-the-art performance among diffusion models on language modeling benchmarks and enables generation of arbitrary-length sequences. We provide the code[1], along with the model weights and blog post on the project page:

https://m-arriola.com/bd3lms

## 1 INTRODUCTION

Diffusion models are widely used to generate images (Ho et al., 2020; Dhariwal & Nichol, 2021; Sahoo et al., 2024b) and videos (Ho et al., 2022; Gupta et al., 2023), and are becoming increasingly effective at generating discrete data such as text (Lou et al., 2024; Sahoo et al., 2024a) or biological sequences (Avdeyev et al., 2023; Goel et al., 2024). Compared to autoregressive models, diffusion models have the potential to accelerate generation and improve the controllability of model outputs (Schiff et al., 2024; Nisonoff et al., 2024; Li et al., 2024; Sahoo et al., 2024c).

Discrete diffusion models currently face at least three limitations. First, in applications such as chat systems, models must generate output sequences of arbitrary length (e.g., a response to a user's question). However, most recent diffusion architectures only generate fixed-length vectors (Austin et al., 2021; Lou et al., 2024). Second, discrete diffusion uses bidirectional context during generation and therefore cannot reuse previous computations with KV caching, which makes inference less efficient (Israel et al., 2025). Third, the quality of discrete diffusion models, as measured by standard metrics such as perplexity, lags behind autoregressive approaches and further limits their applicability (Gulrajani & Hashimoto, 2024; Sahoo et al., 2024a).

This paper makes progress towards addressing these limitations by introducing Block Discrete Denoising Diffusion Language Models (BD3-LMs), which interpolate between discrete diffusion and autoregressive models. Specifically, block diffusion models (also known as semi-autoregressive models) define an autoregressive probability distribution over blocks of discrete random variables (Si et al., 2022; 2023); the conditional probability of a block given previous blocks is specified by a discrete denoising diffusion model (Austin et al., 2021; Sahoo et al., 2024a).

Developing effective BD3-LMs involves two challenges. First, efficiently computing the training objective for a block diffusion model is not possible using one standard forward pass of a neural

---

[*]Correspondence to Marianne Arriola: `marriola@cs.cornell.edu`

[†]Cornell Tech, NY, USA.    [¶]Stanford University, CA, USA.    [‡] Cohere, NY, USA.

[1]Code: https://github.com/kuleshov-group/bd3lms

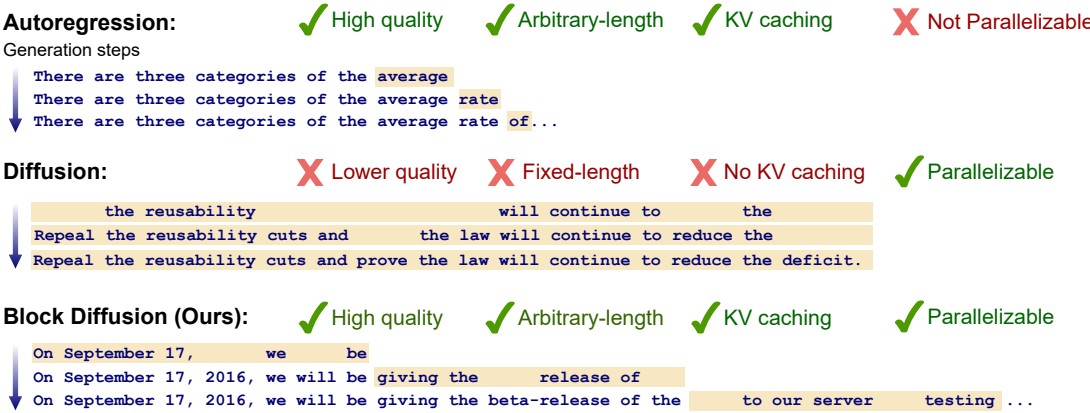

Figure 1: Block diffusion sequentially generates blocks of tokens by performing diffusion within each block and conditioning on previous blocks. By combining strength from autoregressive and diffusion models, block diffusion overcomes the limitations of both approaches by supporting variable-length, higher-quality generation and improving inference efficiency with KV caching and parallel sampling.

network and requires developing specialized algorithms. Second, training is hampered by the high variance of the gradients of the diffusion objective, causing BD3-LMs to under-perform autoregression even with a block size of one (when both models should be equivalent). We derive estimators of gradient variance, and demonstrate that it is a key contributor to the gap in perplexity between autoregression and diffusion. We then propose custom noise processes that minimize gradient variance and make progress towards closing the perplexity gap.

We evaluate BD3-LMs on language modeling benchmarks, and demonstrate that they are able to generate sequences of arbitrary length, including lengths that exceed their training context. In addition, BD3-LMs achieve new state-of-the-art perplexities among discrete diffusion models. Compared to alternative semi-autoregressive formulations that perform Gaussian diffusion over embeddings (Han et al., 2022; 2023), our discrete approach features tractable likelihood estimates and yields samples with improved generative perplexity using an order of magnitude fewer generation steps. In summary, our work makes the following contributions:

- We introduce block discrete diffusion language models, which are autoregressive over blocks of tokens; conditionals over each block are based on discrete diffusion. Unlike prior diffusion models, block diffusion supports variable-length generation and KV caching.

- We introduce custom training algorithms for block diffusion models that enable efficiently leveraging the entire batch of tokens provided to the model.

- We identify gradient variance as a limiting factor of the performance of diffusion models, and we propose custom data-driven noise schedules that reduce gradient variance.

- Our results establish a new state-of-the-art perplexity for discrete diffusion and make progress toward closing the gap to autoregressive models.

## 2 BACKGROUND: LANGUAGE MODELING PARADIGMS

**Notation** We consider scalar discrete random variables with $V$ categories as 'one-hot' column vectors in the space $\mathcal{V} = \{\mathbf{x} \in \{0,1\}^V : \sum_i \mathbf{x}_i = 1\} \subset \Delta^V$ for the simplex $\Delta^V$. Let the $V$-th category denote a special [MASK] token, where $\mathbf{m} \in \mathcal{V}$ is its one-hot vector. We define $\mathbf{x}^{1:L}$ as a sequence of $L$ tokens, where $\mathbf{x}^\ell \in \mathcal{V}$ for all tokens $\ell \in \{1, \ldots, L\}$, and use $\mathcal{V}^L$ to denote the set of all such sequences. Throughout the work, we simplify notation and refer to the token sequence as $\mathbf{x}$ and an individual token as $\mathbf{x}^\ell$. Finally, let $\mathrm{Cat}(\cdot; p)$ be a categorical distribution with probability $p \in \Delta^V$.

## 2.1 Autoregressive Models

Consider a sequence of $L$ tokens $\mathbf{x} = \left[ \mathbf{x}^1, \ldots, \mathbf{x}^L \right]$ drawn from the data distribution $q(\mathbf{x})$. Autoregressive (AR) models define a factorized distribution of the form

$$\log p_\theta(\mathbf{x}) = \sum_{\ell=1}^{L} \log p_\theta(\mathbf{x}^\ell \mid \mathbf{x}^{<\ell}), \tag{1}$$

where each $p_\theta(\mathbf{x}^\ell \mid \mathbf{x}^{<\ell})$ is parameterized directly with a neural network. As a result, AR models may be trained efficiently via next token prediction. However, AR models take $L$ steps to generate $L$ tokens due to the sequential dependencies.

## 2.2 Discrete Denoising Diffusion Probabilistic Models

Diffusion models fit a model $p_\theta(\mathbf{x})$ to reverse a forward corruption process $q$ (Sohl-Dickstein et al., 2015; Ho et al., 2020; Sahoo et al., 2024b). This process starts with clean data $\mathbf{x}$ and defines latent variables $\mathbf{x}_t = \left[ \mathbf{x}_t^1, \ldots, \mathbf{x}_t^L \right]$ for $t \in [0, 1]$, which represent progressively noisier versions of $\mathbf{x}$. Given a discretization into $T$ steps, we define $s(j) = (j-1)/T$ and $t(j) = j/T$. For brevity, we drop $j$ from $t(j)$ and $s(j)$ below; in general, $s$ denotes the time step preceding $t$.

The D3PM framework (Austin et al., 2021) defines $q$ as a Markov forward process acting independently on each token $\mathbf{x}^\ell$: $q(\mathbf{x}_t^\ell \mid \mathbf{x}_s^\ell) = \mathrm{Cat}(\mathbf{x}_t^\ell; Q_t \mathbf{x}_s^\ell)$ where $Q_t \in \mathbb{R}^{V \times V}$ is the diffusion matrix. The matrix $Q_t$ can model various transformations, including masking, random token changes, and related word substitutions.

An ideal diffusion model $p_\theta$ is the reverse of the process $q$. The D3PM framework defines $p_\theta$ as

$$p_\theta(\mathbf{x}_s \mid \mathbf{x}_t) = \prod_{\ell=1}^{L} p_\theta(\mathbf{x}_s^\ell \mid \mathbf{x}_t) = \sum_{\mathbf{x}} \left[ \prod_{\ell=1}^{L} q(\mathbf{x}_s^\ell \mid \mathbf{x}_t^\ell, \mathbf{x}^\ell) p_\theta(\mathbf{x}^\ell \mid \mathbf{x}_t) \right], \tag{2}$$

where the denoising base model $p_\theta(\mathbf{x}^\ell \mid \mathbf{x}_t)$ predicts clean token $\mathbf{x}^\ell$ given the noisy sequence $\mathbf{x}_t$, and the reverse posterior $q(\mathbf{x}_s^\ell \mid \mathbf{x}_t^\ell, \mathbf{x})$ is defined following Austin et al. (2021) in Suppl. B.3.

The diffusion model $p_\theta$ is trained using variational inference. Let $\mathrm{KL}[\cdot]$ denote the Kullback-Leibler divergence. Then, the Negative ELBO (NELBO) is given by (Sohl-Dickstein et al., 2015):

$$\mathcal{L}(\mathbf{x}; \theta) = \mathbb{E}_q \left[ -\log p_\theta(\mathbf{x} | \mathbf{x}_{t(1)}) + \sum_{j=1}^{T} D_{\mathrm{KL}}[q(\mathbf{x}_{s(j)} | \mathbf{x}_{t(j)}, \mathbf{x}) \| p_\theta(\mathbf{x}_{s(j)} | \mathbf{x}_{t(j)})] + D_{\mathrm{KL}}[q(\mathbf{x}_{t(T)} | \mathbf{x}) \| p_\theta(\mathbf{x}_{t(T)})] \right] \tag{3}$$

This formalism extends to continuous time via Markov chain (CTMC) theory and admits score-based generalizations (Song & Ermon, 2019; Lou et al., 2024; Sun et al., 2022). Further simplifications (Sahoo et al., 2024a; Shi et al., 2024; Ou et al., 2025) tighten the ELBO and enhance performance.

## 3 Block Diffusion Language Modeling

We explore a class of Block Discrete Denoising Diffusion Language Models (BD3-LMs) that interpolate between autoregressive and diffusion models by defining an autoregressive distribution over blocks of tokens and performing diffusion within each block. We provide a block diffusion objective for maximum likelihood estimation and efficient training and sampling algorithms. We show that for a block size of one, the diffusion objective suffers from high variance despite being equivalent to the autoregressive likelihood in expectation. We identify high training variance as a limitation of diffusion models and propose data-driven noise schedules that reduce the variance of the gradient updates during training.

### 3.1 Block Diffusion Distributions and Model Architectures

We propose to combine the language modeling paradigms in Sec. 2 by autoregressively modeling blocks of tokens and performing diffusion within each block. We group tokens in $\mathbf{x}$ into $B$ blocks of

length $L'$ with $B = L/L'$ (we assume that $B$ is an integer). We denote each block $\mathbf{x}^{(b-1)L':bL'}$ from token at positions $(b-1)L'$ to $bL'$ for blocks $b \in \{1, \dots, B\}$ as $\mathbf{x}^b$ for simplicity. Our likelihood factorizes over blocks as

$$\log p_\theta(\mathbf{x}) = \sum_{b=1}^{B} \log p_\theta(\mathbf{x}^b \mid \mathbf{x}^{<b}), \tag{4}$$

and each $p_\theta(\mathbf{x}^b \mid \mathbf{x}^{<b})$ is modeled using discrete diffusion over a block of $L'$ tokens. Specifically, we define a reverse diffusion process as in (2), but restricted to block $b$:

$$p_\theta(\mathbf{x}_s^b \mid \mathbf{x}_t^b, \mathbf{x}^{<b}) = \sum_{\mathbf{x}^b} q(\mathbf{x}_s^b \mid \mathbf{x}_t^b, \mathbf{x}^b) p_\theta(\mathbf{x}^b \mid \mathbf{x}_t^b, \mathbf{x}^{<b}) \tag{5}$$

We obtain a principled learning objective by applying the NELBO in (3) to each term in (4) to obtain

$$-\log p_\theta(\mathbf{x}) \leq \mathcal{L}_{\mathrm{BD}}(\mathbf{x}; \theta) := \sum_{b=1}^{B} \mathcal{L}(\mathbf{x}^b, \mathbf{x}^{<b}; \theta), \tag{6}$$

where each $\mathcal{L}(\mathbf{x}^b, \mathbf{x}^{<b}; \theta)$ is an instance of (3) applied to $\log p_\theta(\mathbf{x}^b \mid \mathbf{x}^{<b})$. Since the model is conditioned on $\mathbf{x}^{<b}$, we make the dependence on $\mathbf{x}^{<b}, \theta$ explicit in $\mathcal{L}$. We denote the sum of these terms $\mathcal{L}_{\mathrm{BD}}(\mathbf{x}; \theta)$ (itself a valid NELBO).

**Model Architecture** Crucially, we parameterize the $B$ base denoiser models $p_\theta(\mathbf{x}^b \mid \mathbf{x}_t^b, \mathbf{x}^{<b})$ using a single neural network $\mathbf{x}_\theta$. The neural network $\mathbf{x}_\theta$ outputs not only the probabilities $p_\theta(\mathbf{x}^b \mid \mathbf{x}_t^b, \mathbf{x}^{<b})$, but also computational artifacts for efficient training. This will enable us to compute the loss $\mathcal{L}_{\mathrm{BD}}(\mathbf{x}; \theta)$ in parallel for all $B$ blocks in a memory-efficient manner. Specifically, we parameterize $\mathbf{x}_\theta$ using a transformer (Vaswani et al., 2017) with a block-causal attention mask. The transformer $\mathbf{x}_\theta$ is applied to $L$ tokens, and tokens in block $b$ attend to tokens in blocks 1 to $b$. When $\mathbf{x}_\theta$ is trained, $\mathbf{x}_\theta^b(\mathbf{x}_t^b, \mathbf{x}^{<b})$ yields $L'$ predictions for denoised tokens in block $b$ based on noised $\mathbf{x}_t^b$ and clean $\mathbf{x}^{<b}$.

In autoregressive generation, it is normal to cache keys and values for previously generated tokens to avoid recomputing them at each step. Similarly, we use $\mathbf{K}^b, \mathbf{V}^b$ to denote the keys and values at block $b$, and we define $\mathbf{x}_\theta$ to support these as input and output. The full signature of $\mathbf{x}_\theta$ is

$$\mathbf{x}_{\mathrm{logits}}^b, \mathbf{K}^b, \mathbf{V}^b \leftarrow \mathbf{x}_\theta^b(\mathbf{x}_t^b, \mathbf{K}^{1:b-1}, \mathbf{V}^{1:b-1}) := \mathbf{x}_\theta^b(\mathbf{x}_t^b, \mathbf{x}^{<b}), \tag{7}$$

where $\mathbf{x}_{\mathrm{logits}}^b$ are the predictions for the clean $\mathbf{x}^b$, and $\mathbf{K}^b, \mathbf{V}^b$ is the key-value cache in the forward pass of $\mathbf{x}_\theta$, and $\mathbf{K}^{1:b-1}, \mathbf{V}^{1:b-1}$ are keys and values cached on a forward pass of $\mathbf{x}_\theta$ over $\mathbf{x}^{<b}$ (hence the inputs $\mathbf{x}^{<b}$ and $\mathbf{K}^{1:b-1}, \mathbf{V}^{1:b-1}$ are equivalent).

## 3.2 Efficient Training and Sampling Algorithms

Ideally, we wish to compute the loss $\mathcal{L}_{\mathrm{BD}}(\mathbf{x}; \theta)$ in one forward pass of $\mathbf{x}_\theta$. However, observe that denoising $\mathbf{x}_t^b$ requires a forward pass on this noisy input, while denoising the next blocks requires running $\mathbf{x}_\theta$ on the clean version $\mathbf{x}^b$. Thus every block has to go through the model at least twice.

**Training** Based on this observation, we propose a training algorithm with these minimal computational requirements (Alg. 1). Specifically, we precompute keys and values $\mathbf{K}^{1:B}, \mathbf{V}^{1:B}$ for the full sequence $\mathbf{x}$ in a first forward pass $(\emptyset, \mathbf{K}^{1:B}, \mathbf{V}^{1:B}) \leftarrow \mathbf{x}_\theta(\mathbf{x})$. We then compute denoised predictions for all blocks using $\mathbf{x}_\theta^b(\mathbf{x}_t^b, \mathbf{K}^{1:b-1}, \mathbf{V}^{1:b-1})$. Each token passes through $\mathbf{x}_\theta$ twice.

**Vectorized Training** Naively, we would compute the logits by applying $\mathbf{x}_\theta^b(\mathbf{x}_t^b, \mathbf{K}^{1:b-1}, \mathbf{V}^{1:b-1})$ in a loop $B$ times. We propose a vectorized implementation that computes $\mathcal{L}_{\mathrm{BD}}(\mathbf{x}; \theta)$ in one forward pass on the concatenation $\mathbf{x}_{\mathrm{noisy}} \oplus \mathbf{x}$ of clean data $\mathbf{x}$ with noisy data $\mathbf{x}_{\mathrm{noisy}} = \mathbf{x}_{t_1}^1 \oplus \cdots \oplus \mathbf{x}_{t_B}^B$ obtained by applying a noise level $t_b$ to each block $\mathbf{x}^b$. We design an attention mask for $\mathbf{x}_{\mathrm{noisy}} \oplus \mathbf{x}$ such that noisy tokens attend to other noisy tokens in their block and to all clean tokens in preceding blocks (see Suppl. B.6). Our method keeps the overhead of training BD3-LMs tractable and combines with pretraining to further reduce costs.

**Sampling**  We sample one block at a time, conditioned on previously sampled blocks (Alg 2). We may use any sampling procedure $\textsc{Sample}(\mathbf{x}_\theta^b, \mathbf{K}^{1:b\text{-}1}, \mathbf{V}^{1:b\text{-}1})$ to sample from the conditional distribution $p_\theta(\mathbf{x}_s^b | \mathbf{x}_t^b, \mathbf{x}^{<b})$, where the context conditioning is generated using cross-attention with pre-computed keys and values $\mathbf{K}^{1:b-1}, \mathbf{V}^{1:b-1}$. Similar to AR models, caching the keys and values saves computation instead of recalculating them when sampling a new block.

Notably, our block diffusion decoding algorithm enables us to sample sequences of arbitrary length, whereas diffusion models are restricted to fixed-length generation. Further, our sampler admits parallel generation within each block, whereas AR samplers are constrained to generate token-by-token.

---

**Algorithm 1** Block Diffusion Training

**Input:** datapoint $\mathbf{x}$, # of blocks $B$, forward noise process $q_t(\cdot|\mathbf{x})$, model $\mathbf{x}_\theta$, loss $\mathcal{L}_{\text{BD}}$
**repeat**
  Sample $t_1, \ldots, t_B \sim \mathcal{U}[0,1]$
  $\forall b \in \{1, ..., B\} : \mathbf{x}_{t_b}^b \sim q_{t_b}(\cdot|\mathbf{x}^b)$
  $\emptyset, \mathbf{K}^{1:B}, \mathbf{V}^{1:B} \leftarrow \mathbf{x}_\theta(\mathbf{x})$  ▷ KV cache
  $\forall b: \mathbf{x}_{\text{logit}}^b, \emptyset, \emptyset \leftarrow \mathbf{x}_\theta^b(\mathbf{x}_{t_b}^b, \mathbf{K}^{1:b-1}, \mathbf{V}^{1:b-1})$
  Let $\mathbf{x}_{\text{logit}} \leftarrow \mathbf{x}_{\text{logit}}^1 \oplus \cdots \oplus \mathbf{x}_{\text{logit}}^B$
  Take gradient step on $\nabla_\theta \mathcal{L}_{\text{BD}}(\mathbf{x}_{\text{logit}}; \theta)$
**until** converged

**Algorithm 2** Block Diffusion Sampling

**Input:** # blocks $B$, model $\mathbf{x}_\theta$, diffusion sampling algorithm $\textsc{Sample}$
$\mathbf{x}, \mathbf{K}, \mathbf{V} \leftarrow \emptyset$  ▷ output & KV cache
**for** $b = 1$ to $B$ **do**
  $\mathbf{x}^b \leftarrow \textsc{Sample}(\mathbf{x}_\theta^b, \mathbf{K}^{1:b\text{-}1}, \mathbf{V}^{1:b\text{-}1})$
  $\emptyset, \mathbf{K}^b, \mathbf{V}^b \leftarrow \mathbf{x}_\theta^b(\mathbf{x}^b)$
  $\mathbf{x} \leftarrow \mathbf{x}^{1:b-1} \oplus \mathbf{x}^b$
  $(\mathbf{K}, \mathbf{V}) \leftarrow (\mathbf{K}^{1:b-1} \oplus \mathbf{K}^b, \mathbf{V}^{1:b-1} \oplus \mathbf{V}^b)$
**end for**
**return** $\mathbf{x}$

---

## 4  UNDERSTANDING LIKELIHOOD GAPS BETWEEN DIFFUSION & AR MODELS

### 4.1  MASKED BD3-LMS

The most effective diffusion language models leverage a masking noise process (Austin et al., 2021; Lou et al., 2024; Sahoo et al., 2024a), where tokens are gradually replaced with a special mask token. Here, we introduce masked BD3-LMs, a special class of block diffusion models based on the masked diffusion language modeling framework (Sahoo et al., 2024a; Shi et al., 2024; Ou et al., 2025).

More formally, we adopt a per-token noise process $q(\mathbf{x}_t^\ell | \mathbf{x}^\ell) = \text{Cat}(\mathbf{x}_t^\ell; \alpha_t \mathbf{x}^\ell + (1 - \alpha_t)\mathbf{m})$ for tokens $\ell \in \{1, \ldots, L\}$ where $\mathbf{m}$ is a one-hot encoding of the mask token, and $\alpha_t \in [0, 1]$ is a strictly decreasing function in $t$, with $\alpha_0 = 1$ and $\alpha_1 = 0$. We employ the linear schedule where the probability of masking a token at time $t$ is $1 - \alpha_t$. We adopt the simplified objective from Sahoo et al. (2024a); Shi et al. (2024); Ou et al. (2025) (the full derivation is provided in Suppl. B.3):

$$-\log p_\theta(\mathbf{x}) \le \mathcal{L}_{\text{BD}}(\mathbf{x}; \theta) := \sum_{b=1}^{B} \mathbb{E}_{t \sim [0,1]} \mathbb{E}_q \frac{\alpha_t'}{1 - \alpha_t} \log p_\theta(\mathbf{x}^b | \mathbf{x}_t^b, \mathbf{x}^{<b}) \qquad (8)$$

where $\alpha_t'$ is the instantaneous rate of change of $\alpha_t$ under the continuous-time extension of (3) that takes $T \to \infty$. The NELBO is tight for $L' = 1$ but becomes a looser approximation of the true negative log-likelihood for $L' \to L$ (see Suppl. B.5).

### 4.2  CASE STUDY: SINGLE TOKEN GENERATION

Our block diffusion parameterization (8) is equivalent in expectation to the autoregressive NLL (1) in the limiting case where $L' = 1$ (see Suppl. B.4). Surprisingly, we find a two point perplexity gap between our block diffusion model for $L' = 1$ and AR when training both models on the LM1B dataset.

Although the objectives are equivalent in expectation, we show that the remaining perplexity gap is a result of high training variance. Whereas AR is trained using the cross-entropy of $L$ tokens, our block diffusion model for $L' = 1$ only computes the cross-entropy for masked tokens $\mathbf{x}_t^\ell = \mathbf{m}$ $\forall \ell \in \{1, \ldots L\}$

Table 1: Test perplexities for single-token generation (PPL; $\downarrow$) across 16B tokens on LM1B.

| | PPL ($\downarrow$) |
|---|---|
| AR | **22.88** |
| + random batch size | 24.37 |
| BD3-LM $L' = 1$ | $\le 25.56$ |
| + tuned schedule | **22.88** |

Train Negative Log-Likelihood (NLL) for Single Token Generation on LM1B

Model — BD3-LM (NELBO) — BD3-LM (Tuned schedule) — AR — AR (random batch size)

Figure 2: Train NLLs for modeling the per-token likelihood on LM1B. Models are trained on 16B tokens. Training under the discrete diffusion NELBO, where half of the tokens in a batch are masked on average, has similar training variance to an AR model with a random batch size.

so that $\mathbb{E}_{t\sim\mathcal{U}[0,1]}q(\mathbf{x}_t^\ell = \mathbf{m}|\mathbf{x}^\ell) = 0.5$. Thus, training on the diffusion objective involves estimating loss gradients with 2x fewer tokens and is responsible for higher training variance compared to AR.

To close the likelihood gap, we train a BD3-LM for $L' = 1$ by designing the forward process to fully mask tokens, i.e. $q(\mathbf{x}_t^\ell = \mathbf{m}|\mathbf{x}^\ell) = 1$. Under this schedule, the diffusion objective becomes *equivalent* to the AR objective (Suppl. B.4). In Table 1, we show that training under the block diffusion objective yields the same perplexity as AR training. Empirically, we see that this reduces the variance of the training loss in Figure 2. We verify that tuning the noise schedule reduces the variance of the objective by measuring $\mathrm{Var}_{\mathbf{x},t}[\mathcal{L}_{\mathrm{BD}}(\mathbf{x};\theta)]$ after training on 328M tokens: while training on the NELBO results in a variance of 1.52, training under full masking reduces the variance to 0.11.

## 4.3 DIFFUSION GAP FROM HIGH VARIANCE TRAINING

Next, we formally describe the issue of gradient variance in training diffusion models. Given our empirical observations for single-token generation, we propose an estimator for gradient variance that we use to minimize the variance of diffusion model training for $L' \geq 1$. While the NELBO is invariant to the choice of noise schedule (Suppl. B.3), this invariance does not hold for our Monte Carlo estimator of the loss used during training. As a result, the variance of the estimator and its gradients are dependent on the schedule. First, we express the estimator of the NELBO with a batch size $K$. We denote a batch of sequences as $\mathbf{X} = [\mathbf{x}^{(1)}, \mathbf{x}^{(2)}, \ldots, \mathbf{x}^{(K)}]$, with each $\mathbf{x}^{(k)} \stackrel{\text{iid}}{\sim} q(\mathbf{x})$. We obtain the batch NELBO estimator below, where $t(k, b)$ is sampled in sequence $k$ and block $b$:

$$\mathcal{L}_{\mathrm{BD}}(\mathbf{X};\theta) := l(\mathbf{X};\theta) = \frac{1}{K}\sum_{k=1}^{K}\sum_{b=1}^{B}\frac{\alpha'_{t(k,b)}}{1-\alpha_{t(k,b)}}\log p_\theta\left(\mathbf{x}^{(k),b} \mid \mathbf{x}_{t(k,b)}^{(k),b}, \mathbf{x}^{(k),<b}\right) \quad (9)$$

The variance of the gradient estimator over $M$ batches for each batch $\mathbf{X}^m$ $\forall m \in \{1, \ldots, M\}$ is:

$$\mathrm{Var}_{\mathbf{X},t}[\nabla_\theta l(\mathbf{X};\theta)] \approx \frac{1}{M-1}\sum_{m=1}^{M}\left\|\nabla_\theta l(\mathbf{X}^m;\theta) - \frac{1}{M}\sum_{m=1}^{M}\nabla_\theta l(\mathbf{X}^m;\theta)\right\|_2^2 \quad (10)$$

## 5 LOW-VARIANCE NOISE SCHEDULES FOR BD3-LMS

### 5.1 INTUITION: AVOID EXTREME MASK RATES

We aim to identify schedules that minimize the variance of the gradient estimator and make training most efficient. In a masked setting, we want to mask random numbers of tokens, so that the model

learns to undo varying levels of noise, which is important during sampling. However, if we mask very few tokens, reconstructing them is easy and does not provide useful learning signal. If we mask everything, the optimal reconstruction are the marginals of each token in the data distribution, which is easy to learn, and again is not useful. These extreme masking rates lead to poor high-variance gradients: we want to learn how to clip them via a simple and effective new class of schedules.

## 5.2 CLIPPED SCHEDULES FOR LOW-VARIANCE GRADIENTS

We propose a class of "clipped" noise schedules that sample mask rates $1 - \alpha_t \sim \mathcal{U}[\beta, \omega]$ for $0 \leq \beta, \omega \leq 1$. We argue that from the perspective of deriving Monte Carlo gradient estimates, these schedules are equivalent to a continuous schedule where the mask probability is approximately 0 before the specified range such that $1 - \alpha_{<\beta} \approx \epsilon$ and approximately 1 after the specified range $1 - \alpha_{>\omega} \approx 1 - \epsilon$. Consequently, $\alpha_t'$ is linear within the range: $\alpha_t' \approx 1/(\beta - \omega)$.

## 5.3 DATA-DRIVEN CLIPPED SCHEDULES ACROSS BLOCK SIZES

As the optimal mask rates may differ depending on the block size $L'$, we adaptively learn the schedule during training. While Kingma et al. (2021) perform variance minimization by isolating a variance term using their squared diffusion loss, this strategy is not directly applicable to our variance estimator in Equation 10 since we seek to reduce variance across random batches in addition to random $t_b$.

Instead, we optimize parameters $\beta, \omega$ to directly minimize training variance. To limit the computational burden of the optimization, we use the variance of the estimator of the diffusion ELBO as a proxy for the gradient estimator to optimize $\beta, \omega$: $\min_{\beta,\omega} \text{Var}_{\mathbf{X},t} [\mathcal{L}(\mathbf{X}; \theta, \beta, \omega)]$. We perform a grid search at regular intervals during training to find the optimal $\beta, \omega$ (experimental details in Sec. 6).

In Table 2, we show that variance of the diffusion NELBO is correlated with test perplexity. Under a range of "clipped" noise rate distributions, we find that there exists a unique distribution for each block size $L' \in \{4, 16, 128\}$ that minimizes both the variance of the NELBO and the test perplexity.

Table 2: Perplexities (PPLs; ↓) and variances of the NELBO $\text{Var}_{\mathbf{X},t} [\mathcal{L}_{\text{BD}}(\mathbf{X}; \theta)]$ (Var. NELBO; ↓). Models are trained on LM1B using a linear schedule for 65B tokens, then finetuned for 10B tokens.

| $L'$ | $\mathcal{U}[0, .5]$ | | $\mathcal{U}[.3, .8]$ | | $\mathcal{U}[.5, 1]$ | | $\mathcal{U}[0, 1]$ | |
|---|---|---|---|---|---|---|---|---|
| | PPL | Var. NELBO | PPL | Var. NELBO | PPL | Var. NELBO | PPL | Var. NELBO |
| 128 | **31.72** | **1.03** | 31.78 | 1.35 | 31.92 | 1.83 | 31.78 | 3.80 |
| 16 | 31.27 | 7.90 | **31.19** | **3.62** | 31.29 | 3.63 | 31.33 | 7.39 |
| 4 | 29.23 | 32.68 | 29.37 | 10.39 | **29.16** | **8.28** | 29.23 | 23.65 |

## 6 EXPERIMENTS

We evaluate BD3-LMs across standard language modeling benchmarks and demonstrate their ability to generate arbitrary-length sequences unconditionally. We pre-train a base BD3-LM using the maximum block size $L' = L$ for 850K gradient steps and fine-tune under varying $L'$ for 150K gradient steps on the One Billion Words dataset (LM1B; Chelba et al. (2014)) and OpenWebText (OWT; Gokaslan et al. (2019)). Details on training and inference are provided in Suppl C.

To reduce the variance of training on the diffusion NELBO, we adaptively learn the range of masking rates by optimizing parameters $\beta, \omega$ as described in Section 5.3. In practice, we do so using a grid search during every validation epoch (after ~5K gradient

Table 3: Test perplexities (PPL; ↓) of models trained for 65B tokens on LM1B. Best diffusion value is bolded.

| | PPL (↓) |
|---|---|
| **Autoregressive** | |
| Transformer-X Base (Dai et al., 2019) | 23.5 |
| Transformer (Sahoo et al., 2024a) | 22.83 |
| **Diffusion** | |
| D3PM (absorb) (Austin et al., 2021) | $\leq 82.34$ |
| SEDD (Lou et al., 2024) | $\leq 32.68$ |
| MDLM (Sahoo et al., 2024a) | $\leq 31.78$ |
| **Block diffusion (Ours)** | |
| BD3-LMs $L' = 16$ | $\leq 30.60$ |
| $L' = 8$ | $\leq 29.83$ |
| $L' = 4$ | $\leq \mathbf{28.23}$ |

updates) to identify $\beta, \omega$: $\min_{\beta, \omega} \text{Var}_{\mathbf{X}, t} \left[ \mathcal{L}(\mathbf{X}; \theta, \beta, \omega) \right]$. During evaluation, we report likelihood under uniformly sampled mask rates (8) as in Austin et al. (2021); Sahoo et al. (2024a).

## 6.1 LIKELIHOOD EVALUATION

On LM1B, BD3-LMs outperform all prior diffusion methods in Table 3. Compared to MDLM (Sahoo et al., 2024a), BD3-LMs achieve up to 13% improvement in perplexity. We observe a similar trend on OpenWebText in Table 4.

We also evaluate the ability of BD3-LMs to generalize to unseen datasets in a zero-shot setting, following the benchmark from Radford et al. (2019). We evaluate the likelihood of models trained with OWT on datasets Penn Tree Bank (PTB; (Marcus et al., 1993)), Wikitext (Merity et al., 2016), LM1B, Lambada (Paperno et al., 2016), AG News (Zhang et al., 2015), and Scientific Papers (Pubmed and Arxiv subsets; (Cohan et al., 2018)). In Table 5, BD3-LM achieves the best zero-shot perplexity on Pubmed, surpassing AR, and the best perplexity among diffusion models on Wikitext, LM1B, and AG News.

Table 4: Test perplexities (PPL; ↓) on OWT for models trained for 524B tokens. Best diffusion value is bolded.

|  | PPL ($\downarrow$) |
|---|---|
| AR (Sahoo et al., 2024a) | 17.54 |
| SEDD (Lou et al., 2024) | $\leq 24.10$ |
| MDLM (Sahoo et al., 2024a) | $\leq 22.98$ |
| BD3-LMs $L' = 16$ | $\leq 22.27$ |
| $L' = 8$ | $\leq 21.68$ |
| $L' = 4$ | $\leq \mathbf{20.73}$ |

Table 5: Zero-shot validation perplexities (↓) of models trained for 524B tokens on OWT. All perplexities for diffusion models are upper bounds.

|  | PTB | Wikitext | LM1B | Lambada | AG News | Pubmed | Arxiv |
|---|---|---|---|---|---|---|---|
| AR | **81.07** | **25.32** | **51.14** | 52.13 | **52.11** | 48.59 | 41.22 |
| SEDD | 96.33 | 35.98 | 68.14 | 48.93 | 67.82 | 45.39 | 40.03 |
| MDLM | 90.96 | 33.22 | 64.94 | **48.29** | 62.78 | 43.13 | **37.89** |
| BD3-LM $L' = 4$ | 96.81 | 31.31 | 60.88 | 50.03 | 61.67 | **42.52** | 39.20 |

## 6.2 SAMPLE QUALITY AND VARIABLE-LENGTH SEQUENCE GENERATION

One key drawback of many existing diffusion language models (e.g,. Austin et al. (2021); Lou et al. (2024)) is that they cannot generate full-length sequences that are longer than the length of the output context chosen at training time. The OWT dataset is useful for examining this limitation, as it contains many documents that are longer than the training context length of 1024 tokens.

We record generation length statistics of 500 variable-length samples in Table 6. We continue sampling tokens until an end-of-sequence token [EOS] is generated or sample quality significantly degrades (as measured by sample entropy). BD3-LMs generate sequences up to $\approx 10 \times$ longer than those of SEDD (Lou et al., 2024), which is restricted to the training context size.

Table 6: Generation length statistics from sampling 500 documents from models trained on OWT.

|  | Median # tokens | Max # tokens |
|---|---|---|
| OWT train set | 717 | 131K |
| AR | 4008 | 131K |
| SEDD | 1021 | 1024 |
| BD3-LM $L' = 16$ | 798 | 9982 |

We also examine the sample quality of BD3-LMs through quantitative and qualitative analyses. In Table 7, we generate sequences of lengths $L = 1024, 2048$ and measure their generative perplexity under GPT2-Large. To sample $L = 2048$ tokens from MDLM, we use their block-wise decoding technique (which does not feature block diffusion training as in BD3-LMs).

We also compare to SSD-LM (Han et al., 2022), an alternative block diffusion formulation. Unlike our discrete diffusion framework, SSD-LM uses Gaussian diffusion and does not support likelihood estimation. Further, BD3-LM adopts an efficient sampler from masked diffusion, where the number of generation steps (NFEs) is upper-bounded by $L$ since tokens are never remasked (Sahoo et al., 2024a; Ou et al., 2025). For SSD-LM, we compare sample quality using $T = 1K$ diffusion steps per block, matching their experimental setting (yielding $\geq 40K$ NFEs), and $T = 25$ where NFEs are comparable across methods.

Table 7: Generative perplexity (Gen. PPL; ↓) and number of function evaluations (NFEs; ↓) of 300 samples of lengths $L = 1024, 2048$. All models are trained on OWT. AR, SEDD, MDLM, BD3-LMs use 110M parameters and are trained on 524B tokens, while SSD-LM uses 400M parameters and is pre-trained on 122B tokens. Best diffusion value is bolded. We provide further details in Suppl. C.5.

| | $L = 1024$ | | $L = 2048$ | |
|---|---|---|---|---|
| Model | Gen. PPL | NFEs | Gen. PPL | NFEs |
| AR | 14.1 | 1K | 13.2 | 2K |
| **Diffusion** | | | | |
| SEDD | 52.0 | 1K | – | – |
| MDLM | 46.8 | 1K | 41.3 | 2K |
| **Block Diffusion** | | | | |
| SSD-LM $L' = 25$ | 37.2 | 40K | 35.3 | 80K |
| | 281.3 | 1K | 281.9 | 2K |
| BD3-LMs $L' = 16$ | 33.4 | 1K | 31.5 | 2K |
| $L' = 8$ | 30.4 | 1K | 28.2 | 2K |
| $L' = 4$ | **25.7** | 1K | **23.6** | 2K |

BD3-LMs achieve the best generative perplexities compared to previous diffusion methods. Relative to SSD-LM, our discrete approach yields samples with improved generative perplexity using an order of magnitude fewer generation steps. We also qualitatively examine samples taken from BD3-LM and baselines (AR, MDLM) trained on the OWT dataset; we report samples in Suppl. D. We observe that BD3-LM samples have higher coherence than MDLM samples and approach the quality of AR.

## 6.3 ABLATIONS

We assess the impact of the design choices in our proposed block diffusion recipes, namely 1) selection of the noise schedule and 2) the efficiency improvement of the proposed training algorithm relative to a naive implementation.

### SELECTING NOISE SCHEDULES TO REDUCE TRAINING VARIANCE

Compared to the linear schedule used in Lou et al. (2024); Sahoo et al. (2024a), training under "clipped" noise schedules is the most effective for reducing the training variance which correlates with test perplexity. In Table 8, the ideal "clipped" masking rates, which are optimized during training, are specific to the block size and further motivate our optimization.

Relative to other standard noise schedules (Chang et al., 2022), "clipped" masking achieves the best performance. As heavier masking is effective for the smaller block size $L' = 4$, we compare with logarithmic and square root schedules that also encourage heavy masking. As lighter masking is optimal for $L' = 16$, we compare with square and cosine schedules.

### EFFICIENCY OF TRAINING ALGORITHM

In the BD3-LM training algorithm (Sec. 3.2), we compute $x_{\text{logit}}$ using two options. We may perform two forward passes through the network (precomputing keys and values for the full sequence $x$, then computing denoised predictions), or combine these passes by concatenating the two inputs into the same attention kernel.

We find that a single forward pass is more efficient as we reduce memory bandwidth bottlenecks by leveraging efficient attention kernels (Dao et al., 2022; Dong et al., 2024), see Suppl. B.7. Instead of paying the cost

Table 8: Effect of the noise schedule on likelihood estimation. We finetune BD3-LMs on 3B tokens from LM1B and evaluate on a linear schedule. For clipped schedules, we compare optimal clipping for $L' = 4, 16$.

| Noise schedule | PPL | Var. NELBO |
|---|---|---|
| **L' = 4** | | |
| Clipped | | |
| $\mathcal{U}[0.45, 0.95]$ | **29.21** | **6.24** |
| $\mathcal{U}[0.3, 0.8]$ | 29.38 | 10.33 |
| Linear $\mathcal{U}[0, 1]$ | 30.18 | 23.45 |
| Logarithmic | 30.36 | 23.53 |
| Square root | 31.41 | 26.43 |
| **L' = 16** | | |
| Clipped | | |
| $\mathcal{U}[0.45, 0.95]$ | 31.42 | 3.60 |
| $\mathcal{U}[0.3, 0.8]$ | **31.12** | **3.58** |
| Linear $\mathcal{U}[0, 1]$ | 31.72 | 7.62 |
| Square | 31.43 | 13.03 |
| Cosine | 31.41 | 13.00 |

of two passes through the network, we only pay the cost of a more expensive attention operation. Our vectorized approach has 20-25% speed-up during training relative to performing two forward passes.

# 7 DISCUSSION AND PRIOR WORK

**Comparison to D3PM** Block diffusion builds off D3PM (Austin et al., 2021) and applies it to each autoregressive conditional. We improve over D3PM in three ways: (1) we extend D3PM beyond fixed sequence lengths; (2) we study the perplexity gap of D3PM and AR models, identify gradient variance as a contributor, and design variance-minimizing schedules; (3) we improve over the perplexity of D3PM models. Our work applies to extensions of D3PM (He et al., 2022; Lou et al., 2024) including ones in continuous time (Campbell et al., 2022; Sun et al., 2022).

**Comparison to MDLM** BD3-LMs further make use of the perplexity-enhancing improvements in MDLM (Sahoo et al., 2024a; Shi et al., 2024; Ou et al., 2025). We also build upon MDLM: (1) while Sahoo et al. (2024a) point out that their NELBO is invariant to the noise schedule, we show that the noise schedule has a significant effect on gradient variance; (2) we push the state-of-the-art in perplexity beyond MDLM. Note that our perplexity improvements stem not only from block diffusion, but also from optimized schedules, and could enhance standard MDLM and D3PM models.

**Comparison to Gaussian Diffusion** Alternatively, one may perform diffusion over continuous embeddings of discrete tokens (Li et al., 2022; Dieleman et al., 2022; Chen et al., 2022). This allows using algorithms for continuous data (Song et al., 2020; Ho & Salimans, 2022), but yields worse perplexity (Graves et al., 2023; Gulrajani & Hashimoto, 2024).

**Comparison to Semi-Autoregressive Diffusion** Han et al. (2022; 2023) introduced a block formulation of Gaussian diffusion. BD3-LMs instead extend Austin et al. (2021), and feature: (1) tractable likelihood estimates for principled evaluation; (2) faster generation, as our number of model calls is bounded by the number of generated tokens, while SSD-LM performs orders of magnitude more calls; (3) improved sample quality. AR-Diffusion (Wu et al., 2023) extends SSD-LM with a left-to-right noise schedule; Chen et al. (2025); Ye et al. (2024) apply to decision traces and videos; Hao et al. (2024); Kong et al. (2025) extend to latent reasoning. PARD (Zhao et al., 2024) applies discrete block diffusion to graphs. In contrast, we (1) interpolate between AR/diffusion performance; (2) support KV caching; (3) perform attention within noised blocks, whereas PARD injects new empty blocks.

Autoregressive diffusion models (Hoogeboom et al., 2021b;a) extend any-order AR models (AO-ARMs; Uria et al. (2014)) to support parallel sampling. Zheng et al. (2024) prove equivalence between MDLM and AO-ARM training. Further extensions of ARMs that compete with diffusion include iterative editing (Gu et al., 2019), parallel and speculative decoding (Gu et al., 2017; Santilli et al., 2023; Cai et al., 2024; Gloeckle et al., 2024), consistency training (Kou et al., 2024), guidance (Sanchez et al., 2023), and cross-modal extensions (Liu et al., 2023; Tian et al., 2025).

**Limitations** Training BD3-LMs is more expensive than regular diffusion training. We propose a vectorized algorithm that keeps training speed within <2x of diffusion training speed; in our experiments, we also pre-train with a standard diffusion loss to further reduce the speed gap. Additionally, BD3-LMs generate blocks sequentially, and hence may face the same speed and controllability constraints as AR especially when blocks are small. Their optimal block size is task specific (e.g., larger for greater control). BD3-LMs are subject to inherent limitations of generative models, including hallucinations (Achiam et al., 2023), copyright infringement (Gokaslan et al., 2024), controllability (Schiff et al., 2024; Wang et al., 2023) and harmful outputs (Bai et al., 2022).

# 8 CONCLUSION

This work explores block diffusion and is motivated by two problems with existing discrete diffusion: the need to generate arbitrary-length sequences and the perplexity gap to autoregressive models. We introduce BD3-LMs, which represent a block-wise extension of the D3PM framework (Austin et al., 2021), and leverage a specialized training algorithm and custom noise schedules that further improve performance. We observe that in addition to being able to generate long-form documents, these models also improve perplexity, setting a new state-of-the-art among discrete diffusion models.

ACKNOWLEDGMENTS AND DISCLOSURE OF FUNDING

This work was partially funded by the National Science Foundation under awards DGE-1922551, CAREER awards 2046760 and 2145577, and by the National Institute of Health under award MIRA R35GM151243. Marianne Arriola is supported by a NSF Graduate Research Fellowship under award DGE-2139899 and a Hopper-Dean/Bowers CIS Deans Excellence Fellowship. We thank Databricks MosaicML for providing access to computational resources.

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

CONTENTS

## A   BLOCK DIFFUSION NELBO

Below, we provide the Negative ELBO (NELBO) for the block diffusion parameterization. Recall that the sequence $\mathbf{x}^{1:L} = \left[\mathbf{x}^1, \ldots, \mathbf{x}^L\right]$ is factorized over $B$ blocks, which we refer to as $\mathbf{x}$ for simplicity, drawn from the data distribution $q(\mathbf{x})$. Specifically, we will factorize the likelihood over $B$ blocks of length $L'$, then perform diffusion in each block over $T$ discretization steps. Let $D_{\mathrm{KL}}[\cdot]$ to denote the Kullback-Leibler divergence, $t, s$ be shorthand for $t(i) = i/T$ and $s(i) = (i-1)/T \ \forall i \in [1, T]$. We derive the NELBO as follows:

$$
\begin{aligned}
-\log p_\theta(\mathbf{x}) &= -\sum_{b=1}^{B} \log p_\theta(\mathbf{x}^b | \mathbf{x}^{<b}) \\
&= -\sum_{b=1}^{B} \log \mathbb{E}_q \frac{p_\theta(\mathbf{x}^b_{t(1):t(T)} | \mathbf{x}^{<b})}{q(\mathbf{x}^b_{t(1):t(T)} | \mathbf{x}^b)} \\
&= -\sum_{b=1}^{B} \log \mathbb{E}_q \frac{p_\theta(\mathbf{x}^b_{t(T)} | \mathbf{x}^{<b}) \prod_{i=1}^{T} p_\theta(\mathbf{x}^b_{s(i)} | \mathbf{x}^b_{t(i)}, \mathbf{x}^{<b})}{\prod_{i=1}^{T} q(\mathbf{x}^b_{t(i)} | \mathbf{x}^b_{s(i)})} \\
&\leq \sum_{b=1}^{B} \Bigg[ \underbrace{-\mathbb{E}_q \log p_\theta(\mathbf{x}^b | \mathbf{x}^b_{t=\frac{1}{T}}, \mathbf{x}^{<b})}_{\mathcal{L}_{\text{recons}}} \\
&\quad + \underbrace{\mathbb{E}_{t \in \left\{\frac{2}{T}, \ldots, \frac{T-1}{T}, 1\right\}} \mathbb{E}_q T D_{KL}\left(q(\mathbf{x}^b_s | \mathbf{x}^b_t, \mathbf{x}^b) \,\|\, p_\theta(\mathbf{x}^b_s | \mathbf{x}^b_t, \mathbf{x}^{<b})\right)}_{\mathcal{L}_{\text{diffusion}}} \\
&\quad + \underbrace{D_{KL}\left(q(\mathbf{x}^b_{t=1} | \mathbf{x}^b) \,\|\, p_\theta(\mathbf{x}^b_{t=1})\right)}_{\mathcal{L}_{\text{prior}}} \Bigg]
\end{aligned}
\tag{11}
$$

## B   MASKED BD3-LMS

We explore a specific class of block diffusion models that builds upon the masked diffusion language modeling framework. In particular, we focus on masking diffusion processes introduced by Austin et al. (2021) and derive a simplified NELBO under this framework as proposed by Sahoo et al. (2024a); Shi et al. (2024); Ou et al. (2025).

First, we define the diffusion matrix $Q_t$ for states $i \in \{1, \ldots, V\}$. Consider the noise schedule function $\alpha_t \in [0, 1]$, which is a strictly decreasing function in $t$ satisfying $\alpha_0 = 1$ and $\alpha_1 = 0$. Denote the mask index as $m = V$. The diffusion matrix is defined by Austin et al. (2021) as:

$$
[Q_t]_{ij} = \begin{cases} 1 & \text{if } i = j = m \\ \alpha_t & \text{if } i = j \neq m \\ 1 - \alpha_t & \text{if } j = m, i \neq m \end{cases}
\tag{12}
$$

The diffusion matrix for the forward marginal $Q_{t|s}$ is:

$$[Q_{t|s}]_{ij} = \begin{cases} 1 & \text{if } i = j = m \\ \alpha_{t|s} & \text{if } i = j \neq m \\ 1 - \alpha_{t|s} & \text{if } j = m, i \neq m \end{cases} \tag{13}$$

where $\alpha_{t|s} = \alpha_t / \alpha_s$.

## B.1 FORWARD PROCESS

Under the D3PM framework (Austin et al., 2021), the forward noise process applied independently for each token $\ell \in \{1, \ldots L\}$ is defined using diffusion matrices $Q_t \in \mathbb{R}^{V \times V}$ as

$$q(\mathbf{x}_t^\ell | \mathbf{x}^\ell) = \text{Cat}\left(\mathbf{x}_t^\ell; \overline{Q}_t \mathbf{x}^\ell\right), \quad \text{with} \quad \overline{Q}_{t(i)} = Q_{t(1)} Q_{t(2)} \ldots Q_{t(i)} \tag{14}$$

## B.2 REVERSE PROCESS

Let $Q_{t|s}$ denote the diffusion matrix for the forward marginal. We obtain the reverse posterior $q(\mathbf{x}_s^\ell \mid \mathbf{x}_t^\ell, \mathbf{x}^\ell)$ using the diffusion matrices:

$$q(\mathbf{x}_s^\ell | \mathbf{x}_t^\ell, \mathbf{x}^\ell) = \frac{q(\mathbf{x}_t^\ell | \mathbf{x}_s^\ell, \mathbf{x}^\ell) q(\mathbf{x}_s^\ell | \mathbf{x}^\ell)}{q(\mathbf{x}_t^\ell | \mathbf{x}^\ell)} = \text{Cat}\left(\mathbf{x}_s^\ell; \frac{Q_{t|s} \mathbf{x}_t^\ell \odot Q_s^\top \mathbf{x}^\ell}{(\mathbf{x}_t^\ell)^\top Q_t^\top \mathbf{x}^\ell}\right) \tag{15}$$

where $\odot$ denotes the Hadmard product between two vectors.

## B.3 SIMPLIFIED NELBO FOR MASKED DIFFUSION PROCESSES

Following Sahoo et al. (2024a); Shi et al. (2024); Ou et al. (2025), we simplify the NELBO in the case of masked diffusion processes. Below, we provide the outline of the NELBO derivation; see the full derivation in Sahoo et al. (2024a); Shi et al. (2024); Ou et al. (2025).

We will first focus on simplifying the diffusion loss term $\mathcal{L}_{\text{diffusion}}$ in Eq. 11. We employ the SUBS-parameterization proposed in Sahoo et al. (2024b) which simplifies the denoising model $p_\theta$ for masked diffusion. In particular, we enforce the following constraints on the design of $p_\theta$ by leveraging the fact that there only exists two possible states in the diffusion process $\mathbf{x}_t^\ell \in \{\mathbf{x}^\ell, \mathbf{m}\} \ \forall \ell \in \{1, \ldots, L\}$.

1. **Zero Masking Probabilities**. We set $p_\theta(\mathbf{x}^\ell = \mathbf{m} | \mathbf{x}_t^\ell) = 0$ (as the clean sequence $\mathbf{x}$ doesn't contain masks).

2. **Carry-Over Unmasking**. The true posterior for the case where $\mathbf{x}_t^\ell \neq \mathbf{m}$ is $q(\mathbf{x}_s^\ell = \mathbf{x}_t^\ell | \mathbf{x}_t^\ell \neq \mathbf{m}) = 1$ (if a token is unmasked in the reverse process, it is never remasked). Thus, we simplify the denoising model by setting $p_\theta(\mathbf{x}_s^\ell = \mathbf{x}_t^\ell | \mathbf{x}_t^\ell \neq \mathbf{m}) = 1$.

As a result, we will only approximate the posterior $p_\theta(\mathbf{x}_s^\ell = \mathbf{x}^\ell | \mathbf{x}_t^\ell = \mathbf{m})$. Let $\mathbf{x}^{b,\ell}$ denote a token in the $\ell$-th position in block $b \in \{1, \ldots, B\}$. The diffusion loss term becomes:

$$\mathcal{L}_{\text{diffusion}} = \sum_{b=1}^{B} \mathbb{E}_t \mathbb{E}_q T \left[ D_{\text{KL}} \left[ q(\mathbf{x}_s^b | \mathbf{x}_t^b, \mathbf{x}^b) \| p_\theta(\mathbf{x}_s^b | \mathbf{x}_t^b, \mathbf{x}^{<b}) \right] \right]$$

$$= \sum_{b=1}^{B} \mathbb{E}_t \mathbb{E}_q T \left[ \sum_{\ell=1}^{L'} D_{\text{KL}} \left[ q(\mathbf{x}_s^{b,\ell} | \mathbf{x}_t^{b,\ell}, \mathbf{x}^{b,\ell}) \| p_\theta(\mathbf{x}_s^{b,\ell} | \mathbf{x}_t^b, \mathbf{x}^{<b}) \right] \right]$$

$D_{\text{KL}}$ is simply the discrete-time diffusion loss for the block $b$; hence, from Sahoo et al. (2024a) (Suppl. B.1), we get:

$$= \sum_{b=1}^{B} \mathbb{E}_t \mathbb{E}_q T \left[ \sum_{\ell=1}^{L'} \frac{\alpha_t - \alpha_s}{1 - \alpha_t} \log p_\theta(\mathbf{x}^{b,\ell} \mid \mathbf{x}_t^{b,\ell}, \mathbf{x}^{<b}) \right]$$

$$= \sum_{b=1}^{B} \mathbb{E}_t \mathbb{E}_q T \left[ \frac{\alpha_t - \alpha_s}{1 - \alpha_t} \log p_\theta(\mathbf{x}^b \mid \mathbf{x}_t^b, \mathbf{x}^{<b}) \right] \tag{16}$$

Lastly, we obtain a tighter approximation of the likelihood by taking the diffusion steps $T \to \infty$ (Sahoo et al., 2024a), for which $T(\alpha_t - \alpha_s) = \alpha'_t$:

$$\mathcal{L}_{\text{diffusion}} = \sum_{b=1}^{B} \mathbb{E}_{t \sim [0,1]} \mathbb{E}_q \left[ \frac{\alpha'_t}{1 - \alpha_t} \log p_\theta(\mathbf{x}^b \mid \mathbf{x}_t^b, \mathbf{x}^{<b}) \right] \tag{17}$$

For the continuous time case, Sahoo et al. (2024a) (Suppl. A.2.4) show the reconstruction loss reduces to 0 as $\mathbf{x}_{t(1)}^b \sim \lim_{T \to \infty} \text{Cat}\left(.; \mathbf{x}_{t=\frac{1}{T}}^b\right) = \text{Cat}(.; \mathbf{x}^b)$. Using this, we obtain:

$$
\begin{aligned}
\mathcal{L}_{\text{recons}} &= -\mathbb{E}_q \log p_\theta(\mathbf{x}^b | \mathbf{x}_{t(1)}^b, \mathbf{x}^{<b}) \\
&= -\log p_\theta(\mathbf{x}^b | \mathbf{x}_{t(1)}^b = \mathbf{x}^b, \mathbf{x}^{<b}) \\
&= 0
\end{aligned}
\tag{18}
$$

The prior loss $\mathcal{L}_{\text{prior}} = \text{D}_{KL}\left(q(\mathbf{x}_{t=1}^b | \mathbf{x}^b) \parallel p_\theta(\mathbf{x}_{t=1}^b)\right)$ also reduces to 0 because $\alpha_{t=1} = 0$ which ensures $q(\mathbf{x}_{t=1}^b | \mathbf{x}^b) = \text{Cat}(.; \mathbf{m})$ and $p_\theta(\mathbf{x}_{t=1}^b) = \text{Cat}(.; \mathbf{m})$; see Sahoo et al. (2024a) (Suppl. A.2.4).

Finally, we obtain a simple objective that is a weighted average of cross-entropy terms:

$$\mathcal{L}_{\text{BD}}(\mathbf{x}; \theta) = \sum_{b=1}^{B} \mathbb{E}_{t \sim [0,1]} \mathbb{E}_q \left[ \frac{\alpha'_t}{1 - \alpha_t} \log p_\theta(\mathbf{x}^b \mid \mathbf{x}_t^b, \mathbf{x}^{<b}) \right] \tag{19}$$

The above NELBO is invariant to the choice of noise schedule $\alpha_t$; see Sahoo et al. (2024a) (Suppl. E.1.1).

### B.4 RECOVERING THE NLL FROM THE NELBO FOR SINGLE TOKEN GENERATION

Consider the block diffuson NELBO for a block size of 1 where $L' = 1, B = L$. The block diffusion NELBO is equivalent to the AR NLL when modeling a single token:

$$-\log p(\mathbf{x}) \leq \sum_{b=1}^{L} \mathbb{E}_{t \sim [0,1]} \mathbb{E}_q \left[ \frac{\alpha'_t}{1 - \alpha_t} \log p_\theta(\mathbf{x}^b \mid \mathbf{x}_t^b, \mathbf{x}^{<b}) \right]$$

$$\because \alpha'_t = -1 \text{ and } \alpha_t = 1 - t,$$

$$= -\sum_{b=1}^{L} \mathbb{E}_{t \sim [0,1]} \mathbb{E}_q \left[ \frac{1}{t} \log p_\theta(\mathbf{x}^b \mid \mathbf{x}_t^b, \mathbf{x}^{<b}) \right]$$

$$= -\sum_{b=1}^{L} \mathbb{E}_{t \sim [0,1]} \frac{1}{t} \mathbb{E}_q \left[ \log p_\theta(\mathbf{x}^b \mid \mathbf{x}_t^b, \mathbf{x}^{<b}) \right]$$

Expanding $\mathbb{E}_q[.]$,

$$
\begin{aligned}
= -\sum_{b=1}^{L} \mathbb{E}_{t \sim [0,1]} \frac{1}{t} \Big[ & q(\mathbf{x}_t^b = \mathbf{m} | \mathbf{x}^b) \log p_\theta(\mathbf{x}^b \mid \mathbf{x}_t^b = \mathbf{m}, \mathbf{x}^{<b}) \\
& + q(\mathbf{x}_t^b = \mathbf{x}^b | \mathbf{x}^b) \log p_\theta(\mathbf{x}^b \mid \mathbf{x}_t^b = \mathbf{x}^b, \mathbf{x}^{<b}) \Big]
\end{aligned}
\tag{20}
$$

Recall that our denoising model employs the SUBS-parameterization proposed in Sahoo et al. (2024b). The "carry-over unmasking" property ensures that $\log p_\theta(\mathbf{x}^b \mid \mathbf{x}_t^b = \mathbf{x}^b, \mathbf{x}^{<b}) = 0$, as an unmasked token is simply copied over from from the input of the denoising model to the output. Hence, (20) reduces to following:

$$-\log p_\theta(\mathbf{x}) \leq -\sum_{b=1}^{L} \mathbb{E}_{t \sim [0,1]} \frac{1}{t} q(\mathbf{x}_t^b = \mathbf{m} | \mathbf{x}^b) \log p_\theta(\mathbf{x}^b \mid \mathbf{x}_t^b = \mathbf{m}, \mathbf{x}^{<b})$$

$$\because q(\mathbf{x}_t^b = \mathbf{m} | \mathbf{x}^b) = t, \text{ we get:}$$

$$= -\sum_{b=1}^{L} \mathbb{E}_{t \sim [0,1]} \log p_\theta(\mathbf{x}^b \mid \mathbf{x}_t^b = \mathbf{m}, \mathbf{x}^{<b})$$

$$= -\sum_{b=1}^{L} \log p_\theta(\mathbf{x}^b \mid \mathbf{m}, \mathbf{x}^{<b}) \tag{21}$$

For single-token generation ($L' = 1$) we recover the autoregressive NLL.

## B.5 Tightness of the NELBO

For block sizes $1 \leq K \leq L$, we show that $-\log p(\mathbf{x}) \leq \mathcal{L}_K \leq \mathcal{L}_{K+1}$. Consider $K = 1$, where we recover the autoregressive NLL (see Suppl B.4):

$$\mathcal{L}_1 = \sum_{b=1}^{L} \log \mathbb{E}_{t\sim[0,1]}\mathbb{E}_q \frac{\alpha'_t}{1-\alpha_t} p_\theta(\mathbf{x}^b \mid \mathbf{x}_t^b, \mathbf{x}^{<b})$$

$$= -\sum_{b=1}^{L} \log p_\theta(\mathbf{x}^b \mid \mathbf{m}, \mathbf{x}^{<b}) \tag{22}$$

Consider the ELBO for block size $K = 2$:

$$\mathcal{L}_2 = \sum_{b=1}^{L/2} \log \mathbb{E}_{t\sim[0,1]}\mathbb{E}_q \frac{\alpha'_t}{1-\alpha_t} p_\theta(\mathbf{x}^b \mid \mathbf{x}_t^b, \mathbf{x}^{<b}) \tag{23}$$

We show that $\mathcal{L}_1 \leq \mathcal{L}_2$, and this holds for all $1 \leq K \leq L$ by induction. Let $\mathbf{x}^{b,\ell}$ correspond to the token in position $\ell \in [1, L']$ of block $b$. We derive the below inequality:

$$-\sum_{b=1}^{L} \log p_\theta(\mathbf{x}^b \mid \mathbf{m}, \mathbf{x}^{<b}) = -\sum_{b=1}^{L/2} \log \mathbb{E}_{t\sim[0,1]}\mathbb{E}_q \frac{1}{1-\alpha_t} p_\theta(\mathbf{x}^b \mid \mathbf{x}_t^b, \mathbf{x}^{<b})$$

$$= -\sum_{b=1}^{L/2} \log \mathbb{E}_{t\sim[0,1]}\mathbb{E}_q \prod_{i=1}^{2} \frac{1}{1-\alpha_t} p_\theta(\mathbf{x}^{b,\ell} \mid \mathbf{x}_t^b, \mathbf{x}^{<b})$$

$$= -\sum_{b=1}^{L/2} \log \prod_{i=1}^{2} \mathbb{E}_{t\sim[0,1]}\mathbb{E}_q \frac{1}{1-\alpha_t} p_\theta(\mathbf{x}^{b,\ell} \mid \mathbf{x}_t^b, \mathbf{x}^{<b})$$

$$\leq -\sum_{b=1}^{L/2}\sum_{i=1}^{2} \log \mathbb{E}_{t\sim[0,1]}\mathbb{E}_q \frac{1}{1-\alpha_t} p_\theta(\mathbf{x}^{b,\ell} \mid \mathbf{x}_t^b, \mathbf{x}^{<b}) \tag{24}$$

## B.6 Specialized Attention Masks

We aim to model conditional probabilities $p_\theta(\mathbf{x}^b \mid \mathbf{x}_t^b, \mathbf{x}^{<b})$ for all blocks $b \in [1, B]$ simultaneously by designing an efficient training algorithm with our transformer backbone. However, modeling all $B$ conditonal terms requires processing both the noised sequence $\mathbf{x}_t^b$ and the conditional context $\mathbf{x}^{<b}$ for all $b$.

Rather than calling the denoising network $B$ times, we process both sequences simultaneously by concatenating them $\mathbf{x}_{\text{full}} \leftarrow \mathbf{x}_t \oplus \mathbf{x}$ as input to a transformer. We update this sequence $\mathbf{x}_{\text{full}}$ of length $2L$ tokens using a custom attention mask $\mathcal{M}_{\text{full}} \in \{0,1\}^{2L\times 2L}$ for efficient training.

The full attention mask is comprised of four $L \times L$ smaller attention masks:

$$\mathcal{M}_{\text{full}} = \begin{bmatrix} \mathcal{M}_{BD} & \mathcal{M}_{OBC} \\ \mathbf{0} & \mathcal{M}_{BC} \end{bmatrix}$$

where $\mathcal{M}_{BD}$ and $\mathcal{M}_{OBC}$ are used to update the representation of $\mathbf{x}_t$ and $\mathcal{M}_{BC}$ is used to update the representation of $\mathbf{x}$. We define these masks as follows:

- $\mathcal{M}_{BD}$ (Block-diagonal mask): Self-attention mask within noised blocks $\mathbf{x}_t^b$

$$[\mathcal{M}_{BD}]_{ij} = \begin{cases} 1 & \text{if } i, j \text{ are in the same block} \\ 0 & \text{otherwise} \end{cases}$$

- $\mathcal{M}_{OBC}$ (Offset block-causal mask): Cross-attention to conditional context $\mathbf{x}^{<b}$

$$[\mathcal{M}_{OBC}]_{ij} = \begin{cases} 1 & \text{if } j \text{ belongs in a block before } i \\ 0 & \text{otherwise} \end{cases}$$

- $\mathcal{M}_{BC}$ (Block-causal mask): Attention mask for updating $\mathbf{x}^b$

$$[\mathcal{M}_{BC}]_{ij} = \begin{cases} 1 & \text{if } j \text{ belongs in the same block as } i, \text{ or a block before } i \\ 0 & \text{otherwise} \end{cases}$$

We visualize an example attention mask for $L = 6$ and block size $L' = 2$ in Figure 3.

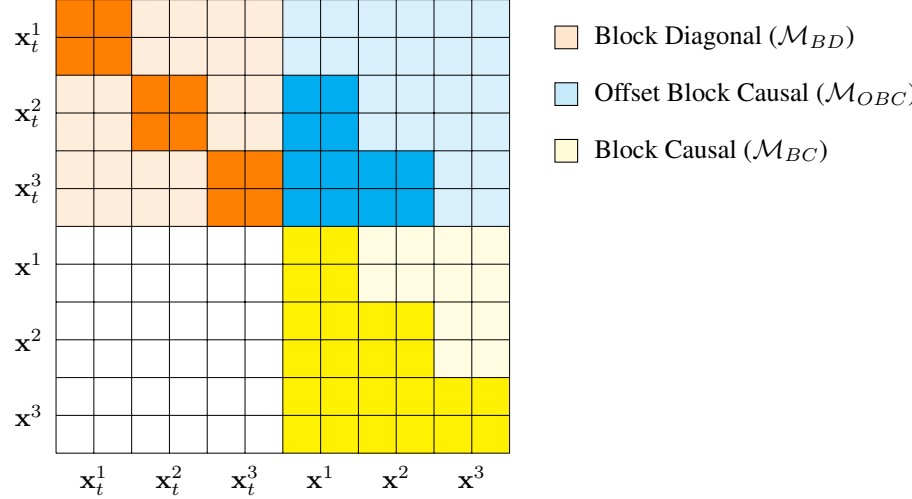

Figure 3: Example of Specialized Attention Mask

### B.7 Optimized Attention Kernel with FlexAttention

As Figure 3 demonstrates, our attention matrix is extremely sparse. We can exploit this sparsity to massively improve the efficiency of BD3-LMs.

FlexAttention (Dong et al., 2024) is a compiler-driven programming model that enables efficient implementation of attention mechanisms with structured sparsity in PyTorch. It provides a flexible interface for defining custom attention masks while maintaining high performance comparable to manually optimized attention kernels.

Below in Fig. 4 we define a block-wise attention mask, `block_diff_mask`, based on its definition as $\mathcal{M}_{\text{full}} \in \{0,1\}^{2L \times 2L}$ in Suppl. B.6. We fuse the attention operations into a single FlexAttention kernel designed to exploit the sparsity in our attention matrix to increase computational efficiency. By doing so, we perform the following optimizations:

- **Precomputed Block Masking:** The `create_block_mask` utility generates a sparse attention mask at compile-time, avoiding per-step computation of invalid attention entries. Through sparsity-aware execution, FlexAttention kernels reduce the number of FLOPs in the attention computation.

- **Reduced Memory Footprint:** By leveraging block-level sparsity, the attention mechanism avoids full materialization of large-scale attention matrices, significantly reducing memory overhead. FlexAttention minimizes memory accesses by skipping fully masked blocks.

- **Optimized Computation via `torch.compile`:** The integration of `torch.compile` enables kernel fusion and efficient execution on GPUs by generating optimized Triton-based kernels. This efficiently parallelizes masked attention computations using optimized GPU execution paths.

```python
def block_diff_mask(b, h, q_idx, kv_idx, block_size, n):
    """
    Constructs the specialized block diffusion attention mask composed of
                                three masks:
    - **Block Diagonal Mask (M_BD)**: Self-attention within noised blocks
    - **Offset Block Causal Mask (M_OBC)**: Cross-attention for
                                conditional context
    - **Block Causal Mask (M_BC)**: Attention to update x0

    Args:
      b, h: Batch and head indices (ignored for mask logic).
      q_idx, kv_idx: Query and Key indices.
      block_size: Defines the block structure.
      n: Sequence length of x_0 and x_t

    Returns:
      A boolean attention mask.
    """

    # Indicate whether token belongs to xt (0) or x0 (1)
    x0_flag_q = (q_idx >= n)
    x0_flag_kv = (kv_idx >= n)

    # Compute block indices
    block_q = torch.where(x0_flag_q == 1,
                          (q_idx - n) // block_size,
                          q_idx // block_size)
    block_kv = torch.where(x0_flag_kv == 1,
                          (kv_idx - n) // block_size,
                          kv_idx // block_size)

    # **1. Block Diagonal Mask (M_BD) **
    block_diagonal = (block_q == block_kv) & (x0_flag_q == x0_flag_kv)

    # **2. Offset Block-Causal Mask (M_OBC) **
    offset_block_causal = (
        (block_q > block_kv)
        & (x0_flag_q == 0)
        & (x0_flag_kv == 1)
    )

    # **3. Block-Causal Mask (M_BC) **
    block_causal = (
        (block_q >= block_kv)
        & (x0_flag_q == 1)
        & (x0_flag_kv == 1)
    )

    # **4. Combine Masks **
    return block_diagonal | offset_block_causal | block_causal
```

Figure 4: We can adapt the masking strategy from Fig. 3 to a FlexAttention compatible sparse masking function as above. This enables the creation of a customized JIT attention operation that uses significantly less memory with up to ≈5X speedup over the naive native scaled_dot_product_attention implementation in PyTorch ($\geq 2.5$) on a A5000 GPU with $L = 1024$ and batch size $B = 16$.

```python
from torch.nn.attention.flex_attention import flex_attention,
                                    create_block_mask
from functools import partial

# Define block-wise attention mask
my_block_diff_mask = partial(block_diff_mask, seq_len=seq_len, block_size
                                    =block_size)

# Generate optimized sparse block mask
block_mask = create_block_mask(my_block_diff_mask, None, None, seq_len*2,
                                    seq_len*2, device=device)

# Compute attention using FlexAttention
# Use no-cudagraphs to avoid an extra copy on small compile graphs.
# Use max-autotune if compiling a larger model all at once.
@torch.compile(fullgraph=True, mode="max-autotune-no-cudagraphs")
def single_pass_block_diff_attn(q, k, v, block_mask):
    return flex_attention(q, k, v, block_mask=block_mask)
```

Figure 5: Attention computation using FlexAttention with our proposed custom mask.

This implementation exploits FlexAttention's ability to dynamically optimize execution based on the provided sparsity pattern. By precomputing block-level sparsity and leveraging efficient kernel fusion, it enables scalable attention computation for long sequences.

Overall, this approach provides a principled method to accelerate attention computations while preserving structured dependency constraints. End-to-end, replacing FlashAttention kernels using a custom mask with FlexAttention kernels leads to $\approx 15\%$ speedup in a model forward pass. We use a single A5000 for $L = 1024$ and batch size $B = 16$.

## C    EXPERIMENTAL DETAILS

We closely follow the same training and evaluation setup as used by Sahoo et al. (2024a).

### C.1    DATASETS

We conduct experiments on two datasets: The One Billion Word Dataset (LM1B; Chelba et al. (2014)) and OpenWebText (OWT; Gokaslan et al. (2019)). Models trained on LM1B use the `bert-base-uncased` tokenizer and a context length of 128. We report perplexities on the test split of LM1B. Models trained on OWT use the `GPT2` tokenizer Radford et al. (2019) and a context length of 1024. Since OWT does not have a validation split, we leave the last 100k documents for validation.

In preparing LM1B examples, Sahoo et al. (2024a) pad each example to fit in the context length of $L = 128$ tokens. Since most examples consist of only a single sentence, block diffusion modeling for larger block sizes $L' > 4$ would not be useful for training. Instead, we concatenate and wrap sequences to a length of 128. As a result, we retrain our autoregressive baseline, SEDD, and MDLM on LM1B with wrapping.

Similarly for OWT, we do not pad or truncate sequences, but concatenate them and wrap them to a length of 1024 similar to LM1B. For unconditional generation experiments in Section 6.2, we wish to generate sequences longer than the context length seen during training. However, Sahoo et al. (2024a) inject beginning-of-sequence and end-of-sequence tokens ([BOS], [EOS] respectively) at the beginning and end of the training context. Thus, baselines from Sahoo et al. (2024a) will generate sequences that match the training context size. To examine model generations across varying lengths in Section 6.2, we retrain our AR, SEDD, and MDLM baselines without injecting [BOS] and [EOS] tokens in the examples. We also adopt this preprocessing convention for training all BD3-LMs on OWT.

### C.2    ARCHITECTURE

The model architecture augments the diffusion transformer (Peebles & Xie, 2023) with rotary positional embeddings (Su et al., 2021). We parameterize our autoregressive baselines, SEDD, MDLM, and BD3-LMs with a transformer architecture from Sahoo et al. (2024a) that uses 12 layers, a hidden dimension of 768, and 12 attention heads. This corresponds to 110M parameters. We do not include timestep conditioning as Sahoo et al. (2024a) show it does not affect performance. We use the AdamW optimizer with a batch size of 512 and constant learning rate warmup from 0 to `3e-4` for 2.5K gradient updates.

### C.3    TRAINING

We train a base BD3-LM using the maximum context length $L' = L$ for 850K gradient steps. Then, we fine-tune under varying $L'$ using the noise schedule optimization for 150K gradient steps on the One Billion Words dataset (LM1B) and OpenWebText (OWT). This translates to 65B tokens and 73 epochs on LM1B, 524B tokens and 60 epochs on OWT. We use 3090, A5000, A6000, and A100 GPUs.

### C.4    LIKELIHOOD EVALUATION

We use a single Monte Carlo estimate for sampling $t$ to evaluate the likelihood of a token block. We adopt a low-discrepancy sampler proposed in Kingma et al. (2021) that reduces the variance of this estimate by ensuring the time steps are more evenly spaced across the interval [0,1] following Sahoo et al. (2024a). In particular, we sample the time step for each block $b \in \{1, \ldots, B\}$ and sequence $k \in \{1, \ldots, K\}$ from a different partition of the uniform interval $t(k,b) \sim \mathcal{U}[\frac{(k-1)B+b-1}{KB}, \frac{(k-1)B+b}{KB}]$.

This low-discrepancy sampler is used for evaluation. For training, each masking probability may be sampled from a "clipped" range $1 - \alpha_t \sim \mathcal{U}[\beta, \omega]$. During training, we uniformly sample $t \in [0, 1]$ under the low-discrepancy sampler. We then apply a linear interpolation to ensure that the masking probability is linear within the desired range: $1 - \alpha_t = \beta + (\omega - \beta)t$.

When reporting zero-shot likelihoods on benchmark datasets from Radford et al. (2019) using models trained on OWT, we wrap all sequences to 1024 tokens and do not add [EOS] between sequences following Sahoo et al. (2024a).

## C.5 INFERENCE

**Generative Perplexity** We report generative perplexity under GPT2-Large from models trained on OWT using a context length of 1024 tokens. Since GPT2-Large uses a context size of 1024, we compute the generative perplexity for samples longer than 1024 tokens using a sliding window with a stride length of 512 tokens.

**Nucleus Sampling** Following SSD-LM (Han et al., 2022), we employ nucleus sampling for BD3-LMs and our baselines. For SSD-LM, we use their default hyperparameters $p = 0.95$ for block size $L' = 25$. For BD3-LMs, AR and MDLM, we use $p = 0.9$. For SEDD, we find that $p = 0.99$ works best.

**Number of Diffusion Steps** In Table 7, BD3-LMs and MDLM use $T = 5K$ diffusion steps. BD3-LMs and MDLM use efficient sampling by caching the output of the denoising network as proposed by Sahoo et al. (2024a); Ou et al. (2025), which ensures that the number of generation steps does not exceed the sample length $L$. Put simply, once a token is unmasked, it is never remasked as a result of the simplified denoising model (Suppl. B.3). We use MDLM's block-wise decoding algorithm for generating variable-length sequences, however these models are not trained with block diffusion. We adopt their default stride length of 512 tokens.

SSD-LM (first row in Table 7) and SEDD use $T = 1K$ diffusion steps. Since block diffusion performs $T$ diffusion steps for each block $b \in \{1, \ldots, B\}$, SSD-LM undergoes $BT$ generation steps. Thus to fairly compare with SSD-LM, we also report generative perplexity for $T = 25$ diffusion steps so that the number of generation steps does not exceed the sequence length (second row in Table 7).

**Improved Categorical Sampling of Diffusion Models** We employ two improvements to Gumbel-based categorical sampling of diffusion models as proposed by Zheng et al. (2024).

First, we use the corrected Gumbel-based categorical sampling from Zheng et al. (2024) by sampling 64-bit Gumbel variables. Reducing the precision to 32-bit has been shown to significantly truncate the Gumbel variables, lowering the temperature and decreasing the sentence entropy.

Second, Zheng et al. (2024) show that the MDLM sampling time scales with the diffusion steps $T$, even though the number of generation steps is bounded by the sequence length. For sample length $L$ and vocabulary size $V$, the sampler requires sampling $\mathcal{O}(TLV)$ uniform variables and performing logarithmic operations on them.

We adopt the first-hitting sampler proposed by Zheng et al. (2024) that requires sampling $\mathcal{O}(LV)$ uniform variables, and thus greatly improves sampling speed especially when $T \gg L$. The first-hitting sampler is theoretically equivalent to the MDLM sampler and leverages two observations: (1) the transition probability is independent of the denoising network, (2) the transition probability is the same for all masked tokens for a given $t$. Thus, the first timestep where a token is unmasked can be analytically sampled as follows (assuming a linear schedule where $\alpha_t = 1 - t$):

$$t_{n-1} = t_n u^{1/n}, \tag{25}$$

where $n \in \{L, \ldots, 1\}$ denotes the number of masked tokens, $u_n \sim \mathcal{U}[0, 1]$ and $t_{n-1}$ corresponds to the first timestep where $n - 1$ tokens are masked.

**Variable-Length Sequence Generation** For arbitrary-length sequence generation using BD3-LMs and AR in Table 6, we continue to sample tokens until the following stopping criteria are met:

1. an [EOS] token is sampled
2. the average entropy of the the last 256-token chunk is below 4

where criterion 2 are necessary to prevent run-on samples from compounding errors (for example, a sequence of repeating tokens). We find that degenerate samples with low entropy result in significantly

low perplexities under `GPT2` and lower the reported generative perplexity. Thus, when a sample meets criterion 2, we regenerate the sample when reporting generative perplexity in Table 7.

## D  SAMPLES

<lendoftextl>'s architect, lawyer and San Giovanni concerto art critic Paolo Capacotti, gained attention from fellow gallery members and even invited him to present a retrospective, publishing issues and newspaper interviews.[10] On 6 September, Kissi and his assistants agreed to move to Angelo's Marcus Collection,[10] which included Giorgio Avolivo Arth and Moscolliso (later owned by the artist Belzina Massingolo) and Pan Giazzoglio Romeam-Guessle. The businessman, Giovanni Paletti, an outstanding collector, owned the museum and the painting. The level of criminal activity around the museum has continued to increase, which is part of several attempts to counter centennial rumors including the possibility that museum staff and visitors are tortured and even exposed to del Cavello for the only full year of Francesco Belzina's life (1999).[4] On the evening of 22 October 2005 it was reported that earlier that evening, guards had come on duty and began flinging an electric field with umbrellas from the balcony. As the fire continued, some of the guards sparked an apparent spat from the window of the cathedral. They remained idly watched by a pile of trash left after a piano key by Pietro Jolla, who died on 21 October 2005.[10] Just before 3:00 to 3pm on Monday, 27 October 2005, strong winds brought the trash on to the residence that opened on 17 October. Some ruined books and statues were hurled in front from every direction of the window. Some claimed that a customer Jacques Monet had beaten the hand of photographer Franco Campetti and in some cases had stuck a broken candle in the doorway of the museum. Andr Romeam-Guessle responded by laughing when he spoke. Giancio Giuliano, the artistic director of the Museum, even tried to told journalists and press that 'the patient in the trisomy machine [sic] carried some corpses four hours into the museum, but the whole time it was the guy who stroked the young man who who broke him'. In 2008, Giuliano told the same press that the hours of the destruction are truly "wrong for their morality" and further stated that 'We are never satisfied with our decision. We made an informed decision to build the museum after destruction.[5] Deaths [ edit ] A little after 12:00 am 17 October 2005, Giuliano and his partner Monica Concerta, noticed that the trash was being thrown by passers-by. Captain Iamienowska leaned over to his film camera and said, in a joking manner, that Iannorello, the chair of the Musceei, was a thief that director Frank Nolan said "he would later be arrested." When Iamienowska arrived, the people in question were interviewed by Captain Anderson Tulaqyuk, a co-man who was initially lying on the scene and whom Iamienowska said was able to stop them from passing in the vicinity. Iguano proceeded to collect the trash and the police arrived, and closed the door of the museum.[6] During the war, the statue structure was partially removed and its cannons damaged. On the eve of the war, the U.S. Army and Canadian Air Company, who once owned a lot on the Coopers of Paris near Leopold Street, sheltered the POWs, who were briefly overworked. This following years led to condemnation of the organization and artists, including John DiDione,[7] returned to the compound to sell the museum items (including many of them originally stolen over the years-more than a million returned); on 12 October, the sculptor Jack Uellein installed a temporary dome on Del Cavella.[10] Among the changes over the years, Diena featured the lion in the middle of the other two halves of the dome and the lion in the back of the Viduccio.[11] In March 2008, Arvnzo Cissino announced that seven studio studios were built between the summer of 1985 and the last one was completed on November 17, 2005.[10] In the fourth quarter of the mid-2000s, a number of new structures were constructed by various museums, but most notably by Arvnzo Cissino, the brothers-soldier, as portrayed by architect Duvois Mayaven in the illusion of the El Valleratas Humanities and Cultural Center, reconstructed by architect Duvois Benjamin. Just as the very significant additions to the current management masts of the del Museum are such private three-bedroom rooms as Tradello's cabin, as well as the architectural aspect from the museum's south wall. Also new are the rockings performed by MC Urquhart, Aiba Ruiz and March Flez, during a music festival held in the US in 2006. Urquhart was later credited with establishing record label Empire Records as an influence on his music. In<lendoftextl>|

Figure 6: Sample from MDLM (Sahoo et al., 2024a) of length $L = 1024$ and $T = 5K$ diffusion steps. The generative perplexity of this sample under GPT2-Large is 69.26 and its entropy is 5.6.

<lendoftextl>When her child was planning to travel to Mexico, she thought it could be easy for her to find money to pay the trip. But that didn't happen, either. The girl arrived on her doorstep the day before after they were scheduled to return to home and - despite her own passport blocked. It wasn't easy. Even her mother reported a bad visit. A gate to the airport- the gate and flight instructions - were locked at nine a.. "I had no way to the gate," she told The Local. "I feel like I should keep my passport and their fees." At first the police arrived, a British man told an e-mail that parents should only have money for themselves to be fine, but could never explain the situation. Later, she told her daughter that her passport had been detained because she wanted to visit her family at a hotel. She felt embarrassed and confused. Nearly two months later, she said, she lost 35(40) on rent and imagined that the traffic-police and customs agents in Bangkok would end up delaying flights and forcing her to stay home. She was worried that her father would refuse allowing her daughter to spend a few days in the country. Meanwhile, the police were sent to search. "It wasn't easy for them when a child feels like home for the first time," said Mahavram Kaas, a spokesman for the Ministry of Foreign Affairs. He's referring to a tour arranged by the French and French foreign ministry, known as Courage in the Child. That tour cost the region 2 million worth of tickets, and cost the Calais family about $25 million in lodging expenses, according to a statement by both the ministry's behalf (he told the Portuguese police agents) and London's Embassy in London (he told the French ambassador they provided a payment for 70,000, which would be used to pay the travel costs for their visits). In 2011, a family from Calais had moved to the UK aged 15. Their sister left to remain in France at 5 years old. Her brother tried to answer that question. He explained it to reporters at his service station at Calais airport. "You take the morning. It's named after you and your little girls," he said. The last mother was having a 19-month stay with her daughter that night, the police said. The first time she was back her husband took their daughter on a boat to the UK, said the mayor of the British Transport Agency. That meant she had plenty of cash-to-go and no money to borrow when she opened an account at Kathmandu Airport a few hours after booking her flight. She panicked. She called the was a Daley's Nessie (small cash register), saying she was getting better. Her doctors visited her when she returned and her boyfriend quit his job for four months after the visit, she said. "How do you feel like you are safe?" A text from a friend left her to the police. "She says I must go get my wallet," said Ajaz. Soon after, she booked a plane ticket to Paris and took a metro train to Calais. One night, a French policeman would knock on the door of a local council building, open the mailman and the phone and tell her she knew that she could not leave at least one week without food. The four months her daughter spent in the UK was exhausting and hard, and it reached the stage where she realized she could barely stay in Britain. "Now no choice but to go home. Then we regret having a daughter," he said. He thought for a minute. "You've broken your heart." "Today, my daughter and my boyfriend decided to stay in this country for over two months," he wrote in an email with his daughter in his hand. "All our flights cancelled and no security. Shame." Caines' family were also put on leave. The French police paid for her car after she rented it, and her female officer used it for the opening ceremony of her press conference in Thailand. The police are still arranging for her family to have their official visit. Although her son is back at work now and his old job, her daughter needs to stay in a hospital in Algiers to continue her education. But, finally, her parents will be making their girl home. The daughter was 18 when they opened her case. She was born two months ago. She doesn't talk about it because it feels like she was still a child, living in Thailand with a small child. Her mother, Anzsa Gurdon, came to England as a three-year-old after her brother, Ehab Rahman, was working as a British worker in Calais while living in London and studying abroad in Dubai. As a mother earns 2,000 pounds a month, they receive a well-paid living in secure accommodation, some even with public transport buses. If they make money, their child stays in the UK, they can set up companies with kids to take care of their children. Other countries sometimes also give birth to parents forced to provide child care. The parents are often refugees from their home countries, they're left without family, and have forced to leave families. As one former refugee fled Syria, his family was in detention, because when and if their child had arrived, they would be living somewhere. The detention centers in Western Europe often have a higher rate for asylum seekers. Often there are higher "safe houses" for young people, and then the people in the center get older when their child comes to stay, but the only family that is a year older is not allowed to have children, and usually only if they stay six months. Children are also detained and are asked to show identification. Because in most cases the refugees ask only about their identity, they don't have access to their own documents, and have no other documentation. An activist working for Ireland says he's against offshore processing. He thinks the charity problem here is like diseases which don't sufficiently seek out international funding. I think too many countries want to employ "humanitarian" children. Now the job Before the refugee crisis in Calais, 1,823 children were living in the UK, the UNHCR website shows. A good chunk of those children had landed in refugee camps in Africa, where mostly African migrants were sending children from Syria and Libya to their camps, but those numbers didn't fare so well. "At the moment when I met the French, it was horrific, they wanted me to put my children in a van. But I was only kidding. It's something called 10-year vans," Aakaz said during an interview. "I want to keep my children for 10 years. That's something. It's like Christmas. The dream of 10 years. . . . For me, the idea that this is a good opportunity, here's a chance," is that she can sleep with her children. At this point, it's much more than just about "alternatives". They now have to decide, at some point, whether they want to take the chance. Is it a big deal or not? Only in Calais She explained what's agreed to so the children can go home and can have a better future. "I'm very determined. My children want to go home but it's my life's personal decision," she said. "Five months. I want them to be home, 5 months. If they're not getting jobs properly, I also want to stay home. But I feel good about what I've got. She is from a poor country. I don't owe anything to anyone. But I have to work for them. I feel like I can just go to the road and provide accommodation for my children and their children." But they all don't work well because their families have her as a head. "They want me to have a job in England, but I feel like it's my home, and I'm not scared of work," she said. "And I feel that the opposite would be possible. I think in the future that I can have a job or two there." A Second World Friday event will be held outside Calais on Saturday, donating 100 euros to the coming week in the money brought up to them by UNHCR through the King Wahab Samba Global Fund. Friends and family expressed their "weakness" like many survivors in many countries."We just don't want to accept what has to happen. We want to put the people back there as soon as possible," said one man. Her brother, who is the son of a long term migrant, said: "The story of the refugee is not a mother's story. The story of the refugee is children's story." "In Calais it's too young for these kids. They play outside or work outside, they just eat, right? I don't think much has changed. This child with all her food and sleep, she's too young for life without any protection. We don't need any protection at all. We need anything that would be safer."<lendoftextl>

Figure 7: Sample from BD3-LM for block size $L' = 16$ of length $L = 2031$ under $T = 5K$ diffusion steps (trained with a context length of $L = 1024$). The generative perplexity of this sample under GPT2-Large is 24.3, and its entropy is 5.5.

<lendoftextl>, but Wilson took over the program and turned it around. "He's done a tremendous job," Caldwell said. "He's done a fantastic job." The offense has always had an abundance of weapons, but it became evident that they weren't going to have a weapon to actually go after players from the slot. Now they're in two different weapons sets. The top group features Dez Bryant and Mohamed Sanu, and the bottom group features an assortment of weapons and pass rushers. The job has become far more complex. The other players can make plays on the ball and get those targets at a higher rate. Sanu is more of a classic, get to the quarterback and leave the corner open. Dontari Poe got the job done this year and became one of the more effective players at the position, even in the passing game. However, Dallas has got to figure out how to get their franchise wideouts to contribute on the field. That can be tough. Adding Poe can help get the receiving corps going. C.J. Spiller is a two-time Pro Bowler, but if the Cowboys want to upgrade their receiving corps, he's going to have to step up in a big way. "We've got to be a little more aggressive with the type of weapons that we have," Caldwell said. "I think that's part of the reason why our last two games, especially when you're playing in Washington, D.C., you've got to be aggressive, make sure you're hitting at every catch. When you are, you're giving up a lot of yards." Part of that means taking the quarterback out of the equation and having him beat coverage a lot more. In the NFC West, you want your offensive weapons to do a better job of running through coverage. The biggest threat that Dallas has is a QB in Ben Roethlisberger. Roethlisberger is far and away the best quarterback in the league, but a lot of the credit has to go to his receiver group. Martavis Bryant and Antonio Brown are both big-time receivers, and last year they were in the top 10 of yards per catch and receiving yards in the league. That production will never be sustainable, but if you're going to be an elite offense, it's going to take a lot of catching up. Roethlisberger is an All-Pro receiver, and he's not the most dynamic option. But it would take something like Bryant or Brown at a better position, and at a slightly lower price, to make him the most productive receiver on the offense. The truth is that Roethlisberger isn't going to be great. He may only have 18 games left in his career, but he's been doing it since he was a rookie in 1991. But that's not the worst thing in the world. Roethlisberger's ability to hit guys on the outside with good movement, vision and running ability is what the Cowboys need in order to keep up with the competition. If he keeps getting better, he could become the best receiver in the league. Follow @walterfootball for updates. Like our Facebook page for more Cowboys news, commentary and conversation.The owner of 1H10 Tree in Charlotte Gardens is taking legal action against the city. Derek Jarman says he's been forced to evict his neighbour, Bob, after he took to social media to threaten to burn down his neighbor's house. "I'm incredibly furious with the city," Jarman told 7.30. "I've been trying to keep my eyes on the prize." Tree in Charlotte Gardens saying it had seen '9,000+ people' enjoying a great weekend The company that owns 1H10 named Bob after a bee and said the tree was frequently targeted because of its unusual location. Bob said he had his concerns about the tree when he was contacted in October. He said they had had 'an ongoing conversation about my neighbor. He called, hung up and he was very threatening' in the 30 days before they turned the tree over to him. A neighbour posted the following online message on 8 October. "I am shocked about the serious problems you are having with your neighbour that has caused you all (sic). You and the 2 of you are making money at the expense of the good people of Charlotte Gardens." Bob says he was furious and said he'd just got off the phone with the city manager. "I told her, 'no, I'm going to bring a lawsuit', and I called the solicitor and tried to get my phone, just hoping the solicitor would help me out. I called again, and I asked if I could go to court and to try and get an injunction. "They told me 'you cannot', and they said, 'we can't, we can't' because you're sending people to the police'." Tree in Charlotte Gardens (Facebook) He also said he'd threatened the city attorney if he didn't stop the building from burning down. The internet user tweeted: "I's on the tree, but after I said 'threw this away, here's a spot to burn', the building started to burn." Tree in Charlotte Gardens (Facebook) Bob said the neighbour had threatened to burn down the tree, the windows, the living room and his entire backyard. "It was more than a threat," he said. "He was a very strong person. He's already damaged so many people in this building. It's not going to go away." Tree in Charlotte Gardens (Facebook) Jarman says he tried to talk the building owner out of the move, but the building owner's behaviour had "deleted him." "I'm going to stop him by letter telling him not to come to my house any more," he said. "I have three kids, and if Bob is going to be in my house, I need to make sure I have someone who can go in there and protect me. "My son does a really good job of protecting me, and I'm not going to let that get in the way of that." Tree in Charlotte Gardens (Facebook) Jarman said Bob had pulled him up on social media, calling him a "white nut" and saying: "For God's sake, stop calling me a white nut. "I should have shut him up on Facebook." He said he sent Bob the letter and thanked him for the support. "He should have done it because he's a real artist and he's a real artist," he said. He declined to name the architect of the new tree, but says the firm is the same one that designs buildings. 'The building is burning down', neighbour says Bob's neighbour, Michael Banks, says the fire is an insult to his daughter. "There are two black women that live next door to me and they told me 'you can't do that', and then the fire went up and then the building burnt down," he said. "You can't burn down a house if you don't burn down the house." Coun-Pete Lawrence, the Northumberland MP for Wood Green, says he has concerns about Bob's neighbours. "It's a very, very sad commentary on the state of society and democracy in general," he said. "It's interesting in a community that's 50,000-plus people, you've got your regular residents and well-meaning neighbours who are apparently oblivious to the destruction of their own home. "To me, that's appalling and it is probably a shocking amount of devastation that it's left behind. "I would expect there to be outrage as well." Bob Jarman fears for his life after the tree was torched Bob says he has told the Northumberland Council that he had already received $1,000 in legal action from the building owner, when he told them about the incident. The building owner has declined to comment on the situation. The builder is currently assessing its legal options. "We've got to sort this out and have an understanding with the builder, Mr Banks," he said. "We've got to make sure we can't get into a legal battle with that person and make that person change his mind. "We don't want to do anything to cause a scene or anybody in the street to be upset." Bob Jarman hopes to have an understanding with the builder on its legal options, who have refused to comment. Topics: state-parliament, smoking-and-doubt, black-wales-6168, united-kingdom, england First postedWhen the other guys are away playing, do a short commercial to get you fired up for the next work day. Once you make it home, get a few junkies for them. They'll be very happy to have you, for at least a day. They might not be so happy after a couple of days. Have a bunch of friends and get ready to keep it going. What are you waiting for? Make this long, one-off<lendoftext>

Figure 8: Sample from an AR model (Sahoo et al., 2024a) with length $L = 2003$ (trained with a context length of $L = 1024$). The generative perplexity of this sample under GPT2-Large is 10.6 and its entropy is 5.5.

