# OpenReview forum: "Block Diffusion: Interpolating Between Autoregressive and Diffusion Language Models"
_ICLR.cc/2025/Conference — ICLR 2025 Oral_

### Official Review · Reviewer_ZiXH · 2024-10-30

**Soundness:** 2
**Presentation:** 3
**Contribution:** 3
**Rating:** 8
**Confidence:** 2

**Summary:**

This paper proposes a semi-autoregressive discrete diffusion model to address the problem that diffusion models cannot perform arbitrary length prediction as autoregressive models.  The paper introduces SAD3-LMs, a class of semi-autoregressive diffusion model, that interpolate between discrete denoising diffusion models and autoregressive models. Compared to existing alternative semi-autoregressive diffusion models with Gaussian diffusion process, the proposed discrete method provides tractable likelihood estimates and yields samples with improved generative perplexity using an order magnitude fewer generation steps.

**Strengths:**

The paper is easy to follow. The paper proposes a semi autoregressive discrete denoising diffusion model and provides an efficient training and sampling algorithm compared to the continuous semi-autoregressive DPM (SSD-LM). The method is incrementally novel. As a semi-autoregressive diffusion model, the paper claimed that the method has advantages in generating full-length sequences that are longer than the length of the output context chosen at training time compared to existing diffusion language models. The paper provides a comprehensive ablation study and well-designed experiments to verify the SAD3-LMs.

**Weaknesses:**

My main concern is the motivation of integrating a discrete diffusion processing in AR. As when $L'=1$, the method performs the same as AR, however, significant higher NFEs, extra tuning required on noise schedules, and complex training and sampling algorithms are involved. The paper can be further improved by showcasing its advantages in real-world applications compared to AR methods.

Moreover, please see Questions for the details.

**Questions:**

1. What is the computational complexity and time complexity for the sampling algorithm given an arbitrary $L'$ and B?

2. If I understand correctly, the SAD3-LMs is trying to address the limitation of diffusion models while maintaining the flexibility of autoregressive models. However, as in Table 7, SAD3-LM has significantly increased NEFs compared to AR, but only with limited PPL improvement. And as in Table 1, when $L'=1$, the SAD3-LM performs exactly the same as AR. What are the advantages of SAD3-LM compared to AR?

3. According to Table 2, the method with different $L'$ values have unique noise rate distributions, this may require extra time and effort for tuning in real applications. I'm curious about can the masked-and-replace diffusion strategy by [1] solve the problem?

4. The paper could be improved by involving an experiment on controlled text generation as in SSD-LM to further showcase its advantages of integrating diffusion models in AR.

Reference

[1] Gu, Shuyang, et al. "Vector quantized diffusion model for text-to-image synthesis." Proceedings of the IEEE/CVF conference on computer vision and pattern recognition. 2022.

---

> ### Author Response · Authors · 2024-11-20
> **Response to ZiXH (1/2)**
>
> We thank the reviewer for their constructive feedback! We address concerns and questions below.
>
> ## **Concern 1**: Identifying Advantages of SAD3-LM relative to AR
> Relative to AR, SAD3-LMs are amenable to faster & controllable sampling and better generalize to out-of-domain data distributions.
>
> ### *Advantage 1: Support for faster generation*
>
> SAD3-LMs have the potential to be faster than AR models because they can output multiple tokens in parallel, which yields an algorithm with improved arithmetic efficiency that better utilizes underlying hardware. We use the number of calls to the transformer (NFEs) as a proxy for inference cost; we see below that the NFEs and sample quality are comparable to AR, and SAD3-LM is much faster compared to previous semi-autoregressive model SSD-LM [1].
> ||Gen. PPL|NFEs|
> |-|-|-|
> |AR|34.04|1024|
> |SSD-LM|36.54|40K|
> |SAD3-LM ($L'=16$)|33.56|1015|
>
> The reviewer is concerned that SAD3-LMs have significantly more NFEs than AR: **we clarify that sampling from SAD3-LMs use *fewer or equal* NFEs as AR**. We use an efficient sampler [2] that caches the predictions of the denoising model until a token is sampled, using at *most* $L$ NFEs. We clarify that SAD3-LM uses 1015 NFEs, and that the previously reported 10K NFEs is a typo (updated in line 431).
>
> A SAD3-LM NFE can oftentimes be run in the same time as an AR NFE by better utilizing FLOPs. While an AR NFE is memory-bound, SAD3-LMs may perform more computation to generate multiple tokens using the same memory transfer. Hence, SAD3-LMs do not incur higher compute costs than AR.
>
> Although we find that SAD3-LMs require the $\text{NFES} \approx L$ to achieve competitive quality, **distillation for discrete diffusion enables sampling 8x faster than AR with improved quality** (concurrent work presented in [3]). We expect SAD3-LMs will be faster than AR after distillation.
>
> ### *Advantage 2: Better generalization to new domains*
>
> **SAD3-LMs better generalize to out-of-domain datasets in a zero-shot setting, whereas AR overfits to the training data.** SAD3-LMs reach better zero-shot likelihoods than AR on out-of-domain datasets Lambada and Scientific Papers (Table 5). SAD3-LM’s discrete diffusion objective prevents overfitting [2]: since the loss gradient is updated only using masked tokens, the noise schedule effectively applies dropout.
>
> ### *Advantage 3: Support for improved controllability*
>
> **SAD3-LM has potential for modular control of samples** similar to SSD-LM by incorporating classifier guidance. Autoregressive models are inherently less amenable to control: they sample a token left-to-right, and a token that has been generated can never change. On the other hand, diffusion models have the potential to edit output tokens multiple times output using a bidirectional context, making it easier to enforce output constraints. While guidance for discrete diffusion is nontrivial and out of the scope of this manuscript (concurrent work presented in [4, 5]), we discuss its potential below (Concern 3).
>
> ---
>
> ## **Concern 2**: “Complex training & sampling algorithms”
> The reviewer is concerned about the complexity of the proposed training and sampling algorithms.
>
> We highlight that our sampler has a simple structure: to sample each block, we may run any diffusion algorithm as a black-box (Alg. 2). To sample subsequent blocks, we simply feed in previously generated blocks as conditional context similar to AR. Thus, our sampling algorithm is as complicated as the chosen diffusion algorithm.
>
> SAD3-LM training is also flexible to the choice in training algorithm: we may use any diffusion backbone to model each block $\mathbf{x}_\theta^b(\mathbf{x}_t^b)$.  However, we find that conditioning the prediction on $\mathbf{x}^{<b}$ in Alg 1 has a significant improvement on likelihoods.
>
> By using a diffusion objective on $\mathbf{x}_t$, SAD3-LMs enjoy benefits of diffusion models such as better generalization and long-term planning without being restricted to purely sequential generation.
>
> ---
>
> ## **Concern 3**: Experiments on controlled text generation
> We agree with the reviewer that controllability is one advantage of diffusion over AR. However, standard guidance techniques are not directly applicable because they require taking gradients with respect to the discrete latents $\mathbf{x}_t$ which are undefined. Although discrete guidance is out of the scope of this manuscript, recent work in this space which applies guidance over transition rate matrices [4] is promising to incorporate in future work. Compared to SSD-LM which performs guidance without principled likelihoods [1], we believe classifier guidance will be better suited for SAD3-LMs because they produce likelihoods that are necessary to compute classifier gradients.

---

> > ### Author Response · Authors · 2024-11-20
> > **Response to ZiXH (2/2)**
> >
> > ## Additional questions
> > > **“What is the computational and time complexity for the sampling algorithm?”**
> >
> > Assume $T$ diffusion steps, $R$ layers, batch size $K$, vocab size $D_{vocab}$, hidden dimension $D_{hid}$,  and sequence length $L = BL’$ for $B$ blocks of length $L’$. Our sampler uses $\approx K * T * R * (4L^2D_{hid} * (\frac{1}{2} + \frac{L’}{2L}) + (2 * L * D_{vocab} * D_{hid}) + (8 L * D_{hid}^2))$ FLOPs and has quadratic time complexity $O(T * L^2)$ similar to AR and diffusion.
> >
> > For an attention matrix of size $L \times L$, AR computes ½ of entries using a causal mask, SAD3-LM computes $\frac{1}{2} + \frac{L’}{2L}$ of the entries using a block-causal mask, and Diffusion LMs compute all $L^2$ entries.
> >
> > > **Can the mask-and-replace diffusion strategy replace noise schedule tuning?**
> >
> > The “mask-and-replace” diffusion strategy improves the posterior prediction in uniform-state diffusion by including an absorbing state [6]. However, the problem of high training variance is caused by uniformly sampling $t$ and is independent of the transition states. For example if $t \approx 0$, the input is mostly clean tokens and denoising task is trivial, so the gradient is not useful for training. We argue that the problem of training variance requires tuning the schedule to avoid extreme noise rates.
> >
> > ---
> > **References:**
> >
> > [1] Han, X, et al. “SSD-LM: Semi-autoregressive Simplex-based Diffusion Language Model for Text Generation and Modular Control.” ACL 2023.
> >
> > [2] Sahoo, S. S.., et al. “Simple and Effective Masked Diffusion Language Models.” NeurIPS 2024.
> >
> > [3] Deschenaux, J., et al. “Beyond Autoregression: Fast LLMs via Self-Distillation Through Time.” arXiv preprint 2024.
> >
> > [4] Nisonoff, H., et al. “Unlocking Guidance for Discrete State-Space Diffusion and Flow Models.” arXiv preprint 2024.
> >
> > [5] Li, X., et al. “Derivative-Free Guidance in Continuous and Discrete Diffusion Models with Soft Value-Based Decoding.” arXiv preprint 2024.
> >
> > [6] Gu, Shuyang, et al. “Vector quantized diffusion model for text-to-image synthesis.” CVPR 2022.

---

> > ### Comment · Reviewer_ZiXH · 2024-11-24
> >
> > Thank the author for the clarification, most of my concerns are addressed.  I have adjusted my score.

---

### Official Review · Reviewer_ix6w · 2024-11-03

**Soundness:** 4
**Presentation:** 4
**Contribution:** 4
**Rating:** 8
**Confidence:** 3

**Summary:**

This paper describes a diffusion language model that is autoregressive between blocks.  By being semi-autoregressive, the model achieves lower perplexity than other diffusion models.

**Strengths:**

The key contribution of this paper is to demonstrate that it is possible to bridge part of the perplexity gap between diffusion-based and autoregressive language models by making a diffusion model partly autoregressive.

The analysis of gradient variance in section 4 is highly appreciated, and will influence the way in which other investigators conduct research.

**Weaknesses:**

The statistical significance of performance differences is not specified.  The differences are large enough to appear statistically significant, but it would be nice to have confirmation.

**Questions:**

p. 3:

Eq. (3) is a negative ELBO.  I think therefore that either Eq. (5) should show -log ptheta(x) <= L_{SAR}, or else it should define L_{SAR} to be the negative of the sum.  If you apply the former solution, I think you need to change the acronym ELBO in the text before and after Eq. (5) to NELBO.

The meaning of Algorithm 1 is obfuscated by the call x_\theta(x), in which the argument, x, has no superscript or subscript.  Is this argument the clean sequence (as in a teacher-forcing mode), or the noisy sequence?  The first paragraph of section 3.2 suggests that x is the clean sequence, in that it specifies two calls to x_\theta: one on the noisy input (which seems to match the line of Algorithm 1 that generates x_{logit}^b), and one on the clean input (which might be the line that generates K^{1:B} and V^{1:B}, but this is the point on which I'm not clear).  On the other hand, Algorithm 2 seems to be defining x, with no superscript or subscript, to be the noisy sequence.

p. 4

based the -> based on the

---

> ### Author Response · Authors · 2024-11-21
> **Response to ix6w**
>
> We thank the reviewer for their feedback! We answer specific questions below.
>
> ## **Concern 1:** Statistical significance of results
> The reviewer asks for comment on the statistical significance of the results. We show that our reported likelihoods are significantly significant by evaluating over three random seeds (for sampling $t$, $\mathbf{x}_t$) on OWT below.
>
> | | SAD3-LM $L'=4$   | SAD3-LM $L'=8$   | SAD3-LM $L'=16$   | MDLM               | AR     |
> |------------|-----------------|-----------------|------------------|--------------------|--------|
> |Test PPL over 3 evaluation seeds | $\leq 20.68$ $\pm 0.01$ | $\leq 21.56$ $\pm 0.01$ | $\leq 22.48$ $\pm 0.02$ | $\leq 23.20$ $\pm 0.02$ | 17.54 |
>
> SAD3-LMs achieve 16% improvement in likelihood from [1], and the best-performing SAD3-LM is within 17% of the AR perplexity.
>
> We expect that the significance of the likelihood gap to AR will continue to improve with further training, as AR training overfits unlike diffusion models (since token masking acts as a form of dropout).
>
> ## Additional questions
>
> The reviewer asks for clarification on the notation in the training and sampling algorithms (Algs 1, 2). The reviewer is correct that in Alg 1, $\mathbf{x}_\theta (\mathbf{x})$ is a forward pass on the clean input to generate keys and values $\mathbf{K}^{1:B}$ and $\mathbf{V}^{1:B}$.
>
> We clarify that during sampling, we may use a diffusion sampling algorithm $\text{Sample}$ as a black-box to generate $\mathbf{x}\_T^b, \mathbf{x}\_{T-1}^b, \dots, \mathbf{x}^b$. Thus, the noisy sequence $\mathbf{x}_t$ is handled internally in $\text{Sample}$ and not included in the rest of the sampling algorithm.
>
> We thank the reviewer for pointing out two typos: we have updated Eq 5 and the surrounding text to show the negative ELBO, and fixed the grammatical error in line 202.
>
> **References**
>
> [1] Sahoo, S. S.., et al. “Simple and Effective Masked Diffusion Language Models.” NeurIPS 2024.

---

### Official Review · Reviewer_HEpL · 2024-11-04

**Soundness:** 3
**Presentation:** 3
**Contribution:** 3
**Rating:** 8
**Confidence:** 4

**Summary:**

In this paper the authors propose a semi-autoregressive model for text generation called SAD3-LM (semi-autoregressive denoising discrete diffusion), which divides the input sequence of length $L$ in $B$ blocks of length $L’$ ($B = L / L’$) and trains a (transformer-implemented) discrete diffusion model (a la D3PM [7], but focusing more on a recent implementation called MDLM [8] that only performs masking) on each block $b$, which is conditioned on the previous blocks $1,…, b-1$. The authors offer an efficient implementation which caches the $K, V$ values of the transformer computations of previous blocks, where training can be parallelized in a single pass (concatenating the clean input $\textbf{x}$ with a noisy version of the input $\textbf{x}_{\text{noisy}}$ and applying a specialised attention mask, and inference proceeds over the blocks sequentially. The authors, focusing on the masking model, additionally propose a theoretical study of the tightness of the NELBO objective varying the block length (showing is tighter when one reduces block length) and show that the semi-autoregressive training increases variance with respect to standard autoregressive training, proposing a way to reduce it by masking in a clipped range of the number of applied mask tokens (which is learned during training). The authors show extensive experimental results comparing with both standard autoregressive models and diffusion models for text data (D3PM [7], SEDD, MDLM [8], SSD-LM) on a wide variety of benchmarks (LM1B, Lambada, AG News, Pubmed, Arxiv) reaching good results in terms of perplexity and number of function evaluations (when comparing to SSD-LM).

References:

- [7] Austin, J., Johnson, D. D., Ho, J., Tarlow, D., & Van Den Berg, R. (2021). Structured denoising diffusion models in discrete state-spaces. Advances in Neural Information Processing Systems, 34, 17981-17993.
- [8] Sahoo, S. S., Arriola, M., Gokaslan, A., Marroquin, E. M., Rush, A. M., Schiff, Y., ... & Kuleshov, V. Simple and Effective Masked Diffusion Language Models. In ICML 2024 Workshop on Efficient and Accessible Foundation Models for Biological Discovery.

**Strengths:**

1. The idea of a semi-autoregressive model, while not new (there is already a rich literature of models combining autoregression with diffusion, while focusing more on continuous diffusion like SSD-LM (see Weakness point 2), in this paper is developed via discrete diffusion. This is a way of proceeding that is very natural, and shares many parallels with novel methods in the literature focusing on block-wise parallel decoding such as [1] (https://aclanthology.org/2023.acl-long.689). It seems clear that a combination of sequential and parallel inference is a good solution, especially for speeding up the models, but this aspect is not much explored in this work (see Weakness point 4). Also the fact that one can generate sentences of arbitrary length is a big improvement over previous models in terms of usability, paired with the ability to estimate likelihoods (which SSD-LM cannot do).
2. Showing the tightness of the NELBO (Appendix C) is an interesting theoretical argument that relates the standard autoregressive objective to discrete diffusion objectives.
3. Good experimental section and good results over previous discrete diffusion approaches. Still one should have done experiments without first pre-training auto-regressively (see Weakness point 3)
4. The paper is written in a good way, having a good structure, not containing errors and being easy to follow. Still there are two sections (Section 4 and 5) in which some parts are not clear (see Weakness point 1), but apart from those the execution is solid.

References:

- [1] Andrea Santilli, Silvio Severino, Emilian Postolache, Valentino Maiorca, Michele Mancusi, Riccardo Marin, and Emanuele Rodola. 2023. Accelerating Transformer Inference for Translation via Parallel Decoding. In Proceedings of the 61st Annual Meeting of the Association for Computational Linguistics (Volume 1: Long Papers), pages 12336–12355, Toronto, Canada. Association for Computational Linguistics.

**Weaknesses:**

1. Some aspects of the work, especially Sections 4 and 5, are poorly defined and demonstrated. For example it is not very clear why the variance of the gradient estimator has the formula in Eq. 9, and also not very clear why extreme masking leads to poor variance. Authors should try to re-formulate these parts in a more sound way.
2. The authors are missing a good number of citations:
    1. First in [1] (https://aclanthology.org/2023.acl-long.689), the parallel GS-Jacobi decoding algorithm is proposed for text autoregressive models and has a similar structure to the approach proposed by the authors: move on blocks from left to right and perform parallel decoding on each block. The difference between the two approaches is that in [1], parallel decoding is done via Jacobi iterations while here instead one uses discrete diffusion. Even more related, a follow-up [5] (https://arxiv.org/abs/2403.00835), trains a consistency-like model [6] (which is a type of diffusion model) to speed-up the Jacobi iterations on each block. I see many parallels between the presented method and these two methods, especially the second. The authors should cite those papers and especially discuss in how they differ with [5].
    2. Additionally, apart from other prominent approaches for discrete autoregressive diffusion like [3] (https://openreview.net/forum?id=Lm8T39vLDTE), there is a rich literature on combining autoregression with continuous diffusion, e.g. [2, 4] (http://proceedings.mlr.press/v139/rasul21a.html, https://openreview.net/forum?id=0EG6qUQ4xE), which the authors barely touch (citing SSD-LM and Diffusion-lm). Please integrate some more references from this literature.
3. Since SAD3-LM is first pre-trained with standard autoregression for 850K steps is it fair to posit it as a semi-autoregressive discrete diffusion model? It can be that it learns representations especially on the pre-training part and the fine-tuning part is more accessory. I think experiments should have been performed also without pre-training as well in order to see how standard autoregressive the pre-training impacts the final model.
4. One advantage of combining sequential autoregression with parallel decoding, as in [1, 5], is to improve inference speed, given that parallel decoding could be done in less steps than $L’$ steps (the number of steps required for autoregressive decoding on such block). I understand that the goal of this work is not much on improving efficiency with respect to autoregressive models since one does 10K steps instead of 1024 steps (Table 7), and I appreciate the reduced inference burden with respect to SSD-LM, but what is the point on doing discrete diffusion if there isn’t any gain on standard autoregression? I agree it is good to have an alternative to standard autoregression, hence all the efforts in the discrete diffusion area, and maybe improved perplexity will be reached in future, but still I think improved efficiency over standard autoregressive should be the main goal to be sought. I do not think authors need to compare with [1, 5] (which do achieve a speed-up, especially in [5]) but at least point and discuss as future work how to target these goals starting from their proposal and maybe improving via the faster models that are researched in diffusion distillation literature (e.g. the already cited consistency models [6]).

References:
- [1] Andrea Santilli, Silvio Severino, Emilian Postolache, Valentino Maiorca, Michele Mancusi, Riccardo Marin, and Emanuele Rodola. 2023. Accelerating Transformer Inference for Translation via Parallel Decoding. In Proceedings of the 61st Annual Meeting of the Association for Computational Linguistics (Volume 1: Long Papers), pages 12336–12355, Toronto, Canada. Association for Computational Linguistics.
- [2] Rasul, K., Seward, C., Schuster, I., & Vollgraf, R. (2021, July). Autoregressive denoising diffusion models for multivariate probabilistic time series forecasting. In International Conference on Machine Learning (pp. 8857-8868). PMLR.
- [3] Hoogeboom, E., Gritsenko, A. A., Bastings, J., Poole, B., van den Berg, R., & Salimans, T. Autoregressive Diffusion Models. In International Conference on Learning Representations.
- [4] Wu, T., Fan, Z., Liu, X., Zheng, H. T., Gong, Y., Shen, Y., ... & Chen, W. (2023, December). AR-DIFFUSION: auto-regressive diffusion model for text generation. In Proceedings of the 37th International Conference on Neural Information Processing Systems (pp. 39957-39974).
- [5] Kou, S., Hu, L., He, Z., Deng, Z., & Zhang, H. CLLMs: Consistency Large Language Models. In Forty-first International Conference on Machine Learning.
- [6] Song, Y., Dhariwal, P., Chen, M., & Sutskever, I. (2023, July). Consistency Models. In International Conference on Machine Learning (pp. 32211-32252). PMLR.

**Questions:**

- Line 212: The authors should define $\alpha’$, in the main body of the paper not only in the appendix.
- Table 5: Any idea why results are so bad on AG News? And why AR is worse than diffusion approaches in 3 settings, different than when evaluating on in-distribution settings? Do the author think that discrete diffusion generalises better on out-of-distribution settings?
- I don’t understand why comparison is SSD-LM is under ablations section. Is SSD-LM the naive implementation?

---

> ### Author Response · Authors · 2024-11-20
> **Response to HePL (1/2)**
>
> We thank the reviewer for their thorough feedback! We address specific concerns below.
>
> ### **Concern 1**: Efficiency improvement over AR should be discussed in future work
> SAD3-LMs have the potential to be faster than AR models because they can output multiple tokens in parallel, which yields an algorithm with improved arithmetic efficiency that better utilizes underlying hardware. We use the number of calls to the transformer (NFEs) as a proxy for inference cost; we see below that the NFEs and sample quality are comparable to AR, and SAD3-LM is much faster compared to previous semi-autoregressive model SSD-LM [1].
> ||Gen. PPL|NFEs|
> |-|-|-|
> |AR|34.04|1024|
> |SSD-LM|36.54|40K|
> |SAD3-LM ($L'=16$)|33.56|1015|
>
> The reviewer is concerned that SAD3-LMs have significantly more NFEs than AR: **we clarify that sampling from SAD3-LMs use *fewer or equal* NFEs as AR**. We use an efficient sampler [2] that caches the predictions of the denoising model until a token is sampled, using at *most* $L$ NFEs. We clarify that SAD3-LM uses 1015 NFEs, and that the previously reported 10K NFEs is a typo (updated in line 431).
>
> A SAD3-LM NFE can oftentimes be run in the same time as an AR NFE by better utilizing FLOPs. While an AR NFE is memory-bound, SAD3-LMs may perform more computation to generate multiple tokens using the same memory transfer. Hence, SAD3-LMs do not incur higher compute costs than AR.
>
> Although we find that SAD3-LMs require the $\text{NFES} \approx L$ to achieve competitive quality, **distillation for discrete diffusion enables sampling 8x faster than AR with improved quality** (concurrent work presented in [3]). We expect SAD3-LMs will be faster than AR after distillation.
>
> ---
>
> ### **Concern 2**: “Experiments should have been performed without pre-training”
> **Pre-training is not strictly necessary.** At the request of the reviewer, we train a SAD3-LM from scratch on LM1B without the pre-training step, and find that it further closes the gap to AR.
> ||PPL (w/ pretraining)|PPL (w/o pretraining)|
> |-|-|-|
> |SAD3-LM $L'$=4|$\leq$ 28.33|$\leq$ 26.99|
> |AR|-|22.83|
>
> We performed the pretraining step in our experiments to train multiple SAD3-LMs for varying block sizes under a limited compute budget.
>
> The reviewer asks if it is fair to train SAD3-LMs from an autoregressive checkpoint: **we clarify that we do *not* initialize from an autoregressive checkpoint,** and instead initialize from a base SAD3-LM with block size $L’=L$ (Line 338). We aim to close the gap from diffusion (equivalent to our base SAD3-LM with $L’=L$) to AR by fine-tuning SAD3-LMs with block size $L’ < L$. We compare all models for the same number of gradient steps.
>
> ---
>
> ### **Concern 3**: Some aspects of Sections 4 & 5 are unclear
> The reviewer notes that aspects of Sections 4 and 5 (the problem of gradient variance in diffusion models) are unclear. We clarify these aspects below.
>
> > **“It is not very clear why the variance of the gradient estimator has the formula in Eq. 9”**
>
> We clarify that the variance of the gradient estimator derived in Eq. 9 is computed by summing the component-wise variances of the gradient vectors across batches. This simple derivation provides an interpretable and computationally efficient way to quantify training variance.
>
> > **“It is not clear why extreme masking leads to poor variance”**
>
> We provide clearer intuition on the effect of extreme masking on variance. There are two extreme cases:
> 1. If we only mask a single token, then the model doesn’t train on most tokens in the sequence. This is equivalent to performing stochastic gradient descent with a smaller batch size, which increases training variance.
> 2. If we mask almost all tokens, then recovering the clean sequence is near impossible, so the optimal solution is to simply predict the marginals of tokens in the data distribution. Since the sequence doesn’t provide a useful training signal, it is also equivalent to reducing the batch size.
>
> CDCD [4], a continuous diffusion framework, draws a similar conclusion. Whereas [4] proposes to maintain a linear relationship between the noise schedule and the uncertainty of model predictions, our problem formulation is more intuitive.

---

> ### Author Response · Authors · 2024-11-20
> **Response to HePL (2/2)**
>
> ### **Concern 4**: Comparison with Jacobi Decoding & Autoregressive Diffusion
> We thank the reviewer for additional references on Jacobi decoding and autoregressive diffusion. Below we provide a comparison which we have included in Section 7 (line 431).
>
> #### **Jacobi Decoding**
> Jacobi decoding [5] is an AR inference technique that iteratively refines a random sequence and supports parallel generation of token blocks. Consistency LLMs [6] extend [5] to include a fine-tuning objective. We highlight key differences:
> - Attention algorithm: [5,6] preserve causal masking from AR whereas SAD3-LMs may leverage more context (attending to tokens within a block and in previous blocks). SAD3-LMs use bidirectional context in the limiting case for $L’=L$
> - Noising process: [5,6] use uniform noise whereas SAD3-LMs use masking. Masking noise has been shown to be superior in language modeling [7]
> - Block-wise generation: [6] denoises the entire sequence, whereas our semi-autoregressive decoding may use clean conditional context $\mathbf{x}^{<b}$ to enhance predictions.
> - Task: SAD3-LMs support unconditional generation whereas [5,6] assume a translation task
>
> #### **Autoregressive diffusion**
> Autoregressive Diffusion Models [8] generalize order-agnostic autoregressive models and discrete diffusion. In contrast, our approach is not order-agnostic as blocks are modeled autoregressively. AR-Diffusion [9] is a variant of SSD-LM [1] that uses a noise schedule that encourages sampling in a left-to-right manner. However, they sacrifice parallelism by assigning unique, per-token timesteps. [10] performs time series forecasting under a diffusion objective conditioned on observations at the previous timestep. However, their framework is autoregressive, so it is not amenable to parallel sampling or controllability.
>
> ---
>
> ## Additional questions
> > **“Why is AR worse on out-of-distribution settings? Why are the results bad on AG News?”**
>
> SAD3-LMs better generalize to out-of-domain datasets in a zero-shot setting, whereas AR overfits to the training data. SAD3-LM’s discrete diffusion objective prevents overfitting [1]: since the loss gradient is updated only using masked tokens, the noise schedule can be seen as applying dropout. We believe that diffusion models perform worse on AG News because they contain shorter sequences (~200 tokens on average), where a purely autoregressive model benefits from capturing short-range dependencies.
>
> > **“The authors should define $\alpha’$. Why is comparison to SSD-LM under ablations?”**
>
> We thank the reviewer for pointing out these details. We included the definition of $\alpha’$ (line 212). We rearranged comparison to SSD-LM to be under Experiments as Section 6.3 (line 409).
>
> ---
> **References:**
>
> [1] Han, X, et al. “SSD-LM: Semi-autoregressive Simplex-based Diffusion Language Model for Text Generation and Modular Control.” ACL 2023.
>
> [2] Sahoo, S. S., et al. “Simple and Effective Masked Diffusion Language Models.” NeurIPS 2024.
>
> [3] Deschenaux, J, et al. “Beyond Autoregression: Fast LLMs via Self-Distillation Through Time.” arXiv preprint 2024.
>
> [4] Dieleman, S, et al. “Continuous diffusion for categorical data.” arXiv preprint 2022.
>
> [5] Santilli, A, et al. “Accelerating Transformer Inference for Translation via Parallel Decoding.” ACL 2023.
>
> [6] Kou, S, et al. “CLLMs: Consistency Large Language Models.” ICML 2024.
>
> [7] Lou, A. “Discrete Diffusion Modeling by Estimating the Ratios of the Data Distribution.” ICML 2024.
>
> [8] Hoogeboom, E, et al. “Autoregressive Diffusion Models.” ICLR 2022.
>
> [9] Wu, T, et al. “AR-Diffusion: Auto-Regressive Diffusion Model for Text Generation.” NeurIPS 2023.
>
> [10] Rasul, K, et al. “Autoregressive Denoising Diffusion Models for Multivariate Probabilistic Time Series Forecasting.” ICML 2021.

---

> > ### Comment · Reviewer_HEpL · 2024-12-03
> >
> > I thank the authors for their detailed answer, the inclusion of the suggested references in the work and for having fixed the problem related to the inference time experiment. As such I increased my score for 6 to 8.

---

### Official Review · Reviewer_Wz2T · 2024-11-04

**Soundness:** 3
**Presentation:** 3
**Contribution:** 3
**Rating:** 8
**Confidence:** 3

**Summary:**

A new generative model on discrete label sequences is proposed: A hybrid between discrete diffusion and auto-regressive model. Specifically, it operates auto-regressively on a block level, and within a block, it uses a diffusion model. This model is called semi-autoregressive denoising discrete diffusion language model (SAD3-LM).

The model is evaluated on standard language modeling benchmarks. Different noise schedules are compared. For the special case of L'=1, the model also requires a special (no-op) noise schedule to be able to recover the performance of the auto-regressive model, due to the grad variance we get from the noise schedule otherwise.

**Strengths:**

- Discrete diffusion is an interesting research direction with potential interesting applications. This work here presents a new way to improve this, allowing to interpolate between auto-regressive and diffusion models by changing the block size.

- Good experiments, good studies.

**Weaknesses:**

- Some details are unclear (but I think these can be solved quite easily).

- Some discussion on the motivation of such models would be nice.

- No speed measurements (for the different cases: Training, sampling, evaluating log probs for given seqs). Also for the vectorized training.

**Questions:**

More general: What is the motivation to use diffusion models for language modeling? Maybe potentially even better PPL than auto-regressive models? We don't see that here. Or better speed? I assume this is the case, but this is not shown here. Or what else? Some discussion on this would be good.

p3 l143: "computational artifacts for efficient training" - unclear what this means.

Table 2, Table 3, Table 4: I think it would be nice to also include L'=1 here for completeness.


Sec 2.2 "The marginalization is tractable" - What exactly do you mean by that? You mean that you can parallelize over the frames in x (L' for SAD3-LM)? But still you can not calculate the full marginalization, or can you? If not, I'm not sure I would call it like that. Maybe better "the approximation of the marginalization is tractable"? And how is this calculated then? You sample from p(x|x_t)? Or take the N-best from p(x|x_t)?


Sec 2.2: You give some examples of what Q_t could be. But what Q_t do you actually use here in this work? How do you compute the q(x_s^b|x_t^b,x^b) exactly? How do you sample from it for the ELBO? I think this should be specified exactly.


In eq 3, you define the loss L via the negative ELBO, but in eq 5, the loss is using the positive ELBO. Thus, something is wrong here. I assume you want to put some negative sign somewhere.


Sec 3.2: What happens when you do just one forward pass on noisy inputs and directly compute the loss that way? I understand that this is not totally correct, as the auto-regressive model part depends on clean input, but it might still work, and maybe even helps for regularization. It would be interesting to try.


Sec 4.1 Masked SAD3-LMs: The title of the section suggests that this is a different model (p), but from the text, I understand that only the noise process, only the q is different, right? Ah, but this effectively then changes also the SAD3-LM itself via equation 2? Please clarify this.




Sec 4.2 "Although the objectives are equivalent in expectation" - can you show this? (maybe just in the appendix)


Sec 4.2: When you show the variance of the grad, I assume this is the L2 norm? You should write that in the formula, or not? (Same also for equation 9.)

Sec 4.2: What about other methods to reduce the gradient variance? E.g. using multiple samples from q for the ELBO, or gradient clipping, or other methods? (Ah, I see that you discuss this further in Sec 5, where you propose some other methods.)

Sec 4.2 / Table 1: How exactly is the PPL for SAD3-LM calculated? This uses equation 5 and equation 3? How is x_{t(0)} sampled from q? What is T? How are the other x_t sampled?


Table 2, Table 6: The PPL, are they exact or just upper bounds?
What kind of hardware is used for training? How many epochs do you train? How long does it take?

**Details Of Ethics Concerns:**

x

---

> ### Author Response · Authors · 2024-11-21
> **Response to Wz2T (1/3)**
>
> We thank the reviewer for their thoughtful feedback! We address specific concerns and questions below.
>
> ## **Concern 1**: Motivation to use diffusion models for language modeling
> Relative to AR, SAD3-LMs are amenable to faster & controllable sampling and better generalize to out-of-domain data distributions.
>
> ### *Advantage 1: Support for faster generation*
>
> SAD3-LMs have the potential to be faster than AR models because they can output multiple tokens in parallel, which yields an algorithm with improved arithmetic efficiency that better utilizes underlying hardware. We use the number of calls to the transformer (NFEs) as a proxy for inference cost; we see below that the NFEs and sample quality are comparable to AR, and SAD3-LM is much faster compared to previous semi-autoregressive model SSD-LM [1].
> ||Gen. PPL|NFEs|
> |-|-|-|
> |AR|34.04|1024|
> |SSD-LM|36.54|40K|
> |SAD3-LM ($L'=16$)|33.56|1015|
>
> The reviewer is concerned that SAD3-LMs have significantly more NFEs than AR: **we clarify that sampling from SAD3-LMs use *fewer or equal* NFEs as AR**. We use an efficient sampler [2] that caches the predictions of the denoising model until a token is sampled, using at *most* $L$ NFEs. We clarify that SAD3-LM uses 1015 NFEs, and that the previously reported 10K NFEs is a typo (updated in line 431).
>
> A SAD3-LM NFE can oftentimes be run in the same time as an AR NFE by better utilizing FLOPs. While an AR NFE is memory-bound, SAD3-LMs may perform more computation to generate multiple tokens using the same memory transfer. Hence, SAD3-LMs do not incur higher compute costs than AR.
>
> Although we find that SAD3-LMs require the $\text{NFES} \approx L$ to achieve competitive quality, **distillation for discrete diffusion enables sampling 8x faster than AR with improved quality** (concurrent work presented in [3]). We expect SAD3-LMs will be faster than AR after distillation.
>
> ### *Advantage 2: Better generalization to new domains*
>
> **SAD3-LMs better generalize to out-of-domain datasets in a zero-shot setting, whereas AR overfits to the training data.** SAD3-LMs reach better zero-shot likelihoods than AR on out-of-domain datasets Lambada and Scientific Papers (Table 5). SAD3-LM’s discrete diffusion objective prevents overfitting [2]: since the loss gradient is updated only using masked tokens, the noise schedule effectively applies dropout.
>
> ### *Advantage 3: Support for improved controllability*
>
> **SAD3-LM has potential for modular control of samples** by incorporating classifier guidance. Autoregressive models are inherently less amenable to control: they sample a token left-to-right, and a token that has been generated can never change. On the other hand, diffusion models have the potential to edit output tokens multiple times output using a bidirectional context, making it easier to enforce output constraints.
>
> **However, standard guidance techniques are not directly applicable** because they require taking gradients with respect to the discrete latents $\mathbf{x}_t$ which are undefined. Although discrete guidance is out of the scope of this manuscript, concurrent work in this space [4,5] are promising to incorporate in future work, such as [4] which applies guidance over transition rate matrices.
>
> ---
>
> ## **Concern 2**:  Speed measurements for training, evaluation, and sampling
> At the request of the reviewer, we provide speed measurements across 12K examples (batch size 32) from LM1B on a single A5000. SAD3-LM uses block size $L'=16$ and SSD-LM [1] uses block size $L'=25$.
> ||SAD3-LM (*vectorized training*)|SAD3-LM (*forward pass per block*)|SSD-LM|MDLM|AR|
> |-|-|-|-|-|-|
> |Train speed (tok/ms $\uparrow$)|20.5|OOM|6.6|32.2|42.7|
> |Eval speed (tok/ms $\uparrow$)|62.0|OOM|11.2|92.4|116.3|
>
> Compared to SSD-LM, SAD3-LM training is ~4x more compute-efficient and 5x more token-efficient by training over the entire sequence, whereas SSD-LM only models one conditional term $p(\mathbf{x}_0^b | \mathbf{x}_t^{<b}, \mathbf{x}_t^b)$ per forward pass. A naive implementation that models each block sequentially (*forward pass per block*) is infeasible.
>
> During sampling, SAD3-LMs use fewer or equal NFEs as AR. For $L$ tokens and $B$ blocks, $B \leq \text{SAD3-LM NFES} \leq L$ and $\text{AR NFEs} = L$. Although we find that SAD3-LMs require the $\text{NFES} \approx L$ to achieve competitive quality, we believe distillation is promising for up to 8x speedup relative to AR (noted in Concern 1: Advantage #1).

---

> ### Author Response · Authors · 2024-11-21
> **Response to Wz2T (2/3)**
>
> ## *Additional questions*
> **Experimental results for $L’=1$**. At the request of the reviewer we provide the test perplexities of SAD3-LM for $L’=1$, finetuning from a base model where $L’=L$ that is pre-trained for 850K gradient steps following our other experiments. SAD3-LM for $L’=1$ is within a point of AR after only 150k finetuning steps on both OWT and LM1B.
> ||SAD3-LM $L'=L$|SAD3-LM $L'=1$|AR|
> |-|-|-|-|
> |PPL ($\downarrow$) on LM1B|32.03|23.69|22.82|
> |PPL ($\downarrow$) on OWT|23.21|17.98|17.54|
>
> We have compared the effect of the schedule on PPL and variance for SAD3-LM $L’=1$ in Table 1 and Line 261.
>
> **“What happens when you do one forward pass on noisy inputs and directly compute the loss that way?”** We find that a single forward pass on the noisy sequence is a harder task and achieves significantly worse likelihoods compared to training on both $\mathbf{x}_t^b$ and conditional context $\mathbf{x}^{<b}$ (Alg 1). We train a model on $\mathbf{x}_t$ using a single block-causal forward pass. We report the following test perplexities from performing 50K gradient steps on LM1B for block size $L'=4$:
> ||SAD3-LM (train on $\mathbf{x}_t, \mathbf{x}$)|SAD3-LM (train on $\mathbf{x}_t$ only)|
> |-|-|-|
> |PPL ($\downarrow$)|29.2|73.2|
>
> **Likelihood calculation: forward process**
>
> > What is T?
>
> We define $T$ in Lines 82 & 93 as the number of time steps in the forward diffusion process. We tighten the likelihood bound by taking $T \rightarrow \infty$.
>
> > How is $x_{t(0)}$ sampled?
>
> We clarify that $\mathbf{x}_{t(0)}$ is the first latent from the forward noise process: since $T \rightarrow \infty$, the first diffusion step is $t(0)$. This is exactly equal to the sample from the data distribution $\mathbf{x} \sim q$ so the reconstruction term in Eq. 3 evaluates to 0.
>
> > How are other $x_t$ sampled?
>
> We show in line 207 that other $\mathbf{x}_t$ is sampled by randomly masking tokens with probability $1 - \alpha_t$ [2].
>
> > What $Q_t$ is used?
>
> Since we can sample $\mathbf{x}_t$ by randomly masking tokens, materializing a transition matrix $Q_t$ is unnecessary and inefficient. We include the equivalent forward process using this $Q_t$ in Line 82 to show how our diffusion process relates with D3PM [6].
> The form of $Q_t$ for absorbing state diffusion is:
>
> $$
> [Q_t]_{ij} =
> \\begin{cases}
> 1 & \\text{if } i = j = m, \\\\
> \\alpha_t & \\text{if } i = j \\neq m, \\\\
> 1 - \\alpha_t & \\text{if } j = m, i \\neq m.
> \\end{cases}
> $$
>
> **Connecting the generic discrete diffusion ELBO to the SAD3-LM ELBO.** We clarify that SAD3-LM is a meta-algorithm that uses a diffusion algorithm as a black-box. It leverages that algorithm's noising process $q$ and applies it in each block $\mathbf{x}^b$ to define a new probabilistic model. It also leverages the diffusion algorithm's learning and sampling processes. In SAD3-LMs, we use the discrete diffusion model from [2] to model a block of tokens. We have provided the full derivation of the ELBO from Eq 3 to Eq 7 under this diffusion model in Suppl A.
>
> **How do you compute and sample from the posterior?** We clarify that the reverse posterior (Eq 2) is intractable (without assuming tokens are modeled independently) because we must marginalize over all possible sequences $\mathbf{x}$. This summation involves an exponential number of terms, making it computationally infeasible in high dimensions.
>
> Since noise is applied to tokens independently and we model clean tokens independently, we make the posterior tractable by modeling it as:
>
> $p\_{\theta} (\mathbf{x}\_s \mid \mathbf{x}\_t) = \prod_i q(x^i\_s \mid x^i\_t, x^i) p\_{\theta} (x^i \mid \mathbf{x}\_t)$
>
> During training, we may compute the posterior probabilities in the ELBO as in Eq 7 which does not require sampling. At inference, we sample $\mathbf{x}\_s \sim p\_{\theta} (\mathbf{x}\_s | \mathbf{x}_t)$ using Gumbel-Max sampling as in standard language modeling.
>
> **Show that the AR and SAD3-LM $L’=1$ objectives are equivalent?** We show their equivalence updated in Suppl B.
>
> **How is perplexity calculated?** We calculate the PPL for SAD3-LM as follows for a sequence $\mathbf{x} = (x^1, \dots, x^L)$ and $B$ blocks using the ELBO in Eq 5:
> $$\text{PPL}(\mathbf{x}) = \text{exp} \Bigg\\{ - \frac{1}{L} \sum_{b=1}^B \mathcal{L} (\mathbf{x}^b, \mathbf{x}^{<b}, \theta) \Bigg\\}  \$$

---

> ### Author Response · Authors · 2024-11-21
> **Response to Wz2T (3/3)**
>
> **Does the variance of the gradient use the L2 Norm?** Yes, we use the L2 norm for the variance of the gradient in Eq 9. The variance of the gradient estimator derived in Eq. 9 is computed by summing the component-wise variances of the gradient vectors across batches.
>
> **What about other methods to reduce the gradient variance?** Besides our noise schedule optimization, we agree with the reviewer that other strategies may be useful. Our training algorithm already implements gradient clipping following [7], yet we still find that training variance still affects performance. Drawing multiple samples from $q(\mathbf{x}_t | \mathbf{x})$ may also be effective yet is costly to perform during training.
>
> **What is meant by “computational artifacts for efficient training”?** We clarify that “computational artifacts for efficient training” in line 143 refers to the precomputed keys and values used for denoising a block conditioned on previous blocks (Line 163).
>
> **Training details?** We have updated training details in the bottom of Suppl F.
>
> **Missing sign in the negative ELBO.** We thank the reviewer for pointing out this detail, which is updated in Eq 5.
>
> ---
>
> **References:**
>
> [1] Han, X, et al. “SSD-LM: Semi-autoregressive Simplex-based Diffusion Language Model for Text Generation and Modular Control.” ACL 2023.
>
> [2] Sahoo, S. S.., et al. “Simple and Effective Masked Diffusion Language Models.” NeurIPS 2024.
>
> [3] Deschenaux, J., et al. “Beyond Autoregression: Fast LLMs via Self-Distillation Through Time.” arXiv preprint 2024.
>
> [4] Nisonoff, H., et al. “Unlocking Guidance for Discrete State-Space Diffusion and Flow Models.” arXiv preprint 2024.
>
> [5] Li, X., et al. “Derivative-Free Guidance in Continuous and Discrete Diffusion Models with Soft Value-Based Decoding.” arXiv preprint 2024.
>
> [6] Austin, J, et al. “Structured Denoising Diffusion Models in Discrete State-Spaces.” NeurIPS 2021.

---

### Public Comment · ~Lingxiao_Zhao1 · 2025-03-15
**Congrats to the acceptance**

First congrats to the acceptance of the paper. I would like to mention that the exploration of AR + diffusion for block-wise diffusion was first explored in https://arxiv.org/pdf/2402.03687, given that the main idea is overlapped, hope you could mention our work in your related work.

---

> ### Public Comment · ~Marianne_Arriola1 · 2025-03-16
> **Reference added**
>
> Hello, thank you for bringing this to our attention! We have included a citation to your work in the final camera-ready version, we will also update our arXiv paper accordingly
>
> We acknowledge SSD-LM [1] as one of the earlier works studying block-autoregressive diffusion.
>
> We note that while our approaches share conceptual similarity applied to different domains (text vs graphs), BD3-LMs focus on interpolating between AR/diffusion performance by tuning the block size. BD3-LMs also feature KV caching to further improve inference efficiency. Additionally, our training algorithm employs bidirectional attention within noised blocks containing both masked and unmasked tokens.
>
> Thank you again, we appreciate your engagement with our paper.
>
> [1] Han, X, et al. “SSD-LM: Semi-autoregressive Simplex-based Diffusion Language Model for Text Generation and Modular Control.” ACL 2023.

---

### Meta-Review · Area_Chair_UhU9 · 2024-12-18

**Metareview:**

The paper proposes an approach to decompose a sequence into blocks of tokens, within each discrete diffusion is used.

I recommend not only an acceptance but also an oral presentation because the reviewers unanimously agree that the work is theoretically interesting and practically useful. It is clear that great care is given to both the execution of the experiments and the writing.

**Additional Comments On Reviewer Discussion:**

The discussion had been smooth-sailing and healthy. The authors were able to accurately address the concerns, and the reviewers were satisfied with the additional results.

---

### Decision · Program_Chairs · 2025-01-22

Accept (Oral)